# The secreted protein Amuc_1409 from *Akkermansia muciniphila* improves gut health through intestinal stem cell regulation

Eun-Jung Kang [1,2,17], Jae-Hoon Kim[1,3,17], Young Eun Kim[4,5,17], Hana Lee [6], Kwang Bo Jung [6], Dong-Ho Chang[7], Youngjin Lee[7], Shinhye Park[7], Eun-Young Lee [7], Eun-Ji Lee[8], Ho Bum Kang[9], Moon-Young Rhyoo[10], Seungwoo Seo [5], Sohee Park[6,11], Yubin Huh[6,11], Jun Go[1], Jung Hyeon Choi[1], Young-Keun Choi[1], In-Bok Lee[1], Dong-Hee Choi[1], Yun Jeong Seo[1], Jung-Ran Noh[1], Kyoung-Shim Kim [1,12], Jung Hwan Hwang[1,12], Ji-Seon Jeong [4,13], Ha-Jeong Kwon [4], Hee Min Yoo [4,13], Mi-Young Son [6,11], Yeon-Gu Kim[8,14], Dae-Hee Lee [9,15], Tae-Young Kim [5], Hyo-Jung Kwon [2], Myung Hee Kim [7], Byoung-Chan Kim[7,16], Yong-Hoon Kim [1,12] ✉, Dukjin Kang [4] ✉ & Chul-Ho Lee [1,12] ✉

*Akkermansia muciniphila* has received great attention because of its beneficial roles in gut health by regulating gut immunity, promoting intestinal epithelial development, and improving barrier integrity. However, *A. muciniphila*-derived functional molecules regulating gut health are not well understood. Microbiome-secreted proteins act as key arbitrators of host-microbiome crosstalk through interactions with host cells in the gut and are important for understanding host-microbiome relationships. Herein, we report the biological function of Amuc_1409, a previously uncharacterised *A. muciniphila*-secreted protein. Amuc_1409 increased intestinal stem cell (ISC) proliferation and regeneration in ex vivo intestinal organoids and in vivo models of radiation- or chemotherapeutic drug-induced intestinal injury and natural aging with male mice. Mechanistically, Amuc_1409 promoted E-cadherin/β-catenin complex dissociation via interaction with E-cadherin, resulting in the activation of Wnt/β-catenin signaling. Our results demonstrate that Amuc_1409 plays a crucial role in intestinal homeostasis by regulating ISC activity in an E-cadherin-dependent manner and is a promising biomolecule for improving and maintaining gut health.

The gut microbiota has been gaining increasing attention as an important contributor to maintaining host health by regulating host metabolism and immunity[1,2]. Communication between the host and gut microbiota is mediated by microbiota-derived bioactive molecules, such as surface proteins, secreted proteins, metabolites, and extracellular vesicles (EV)[3]. In particular, extracellular secreted

proteins play key mediators in host-microbiome crosstalk, directly interacting with host cells by passing through the mucin layer, thereby modulating host signaling to ensure homeostatic balance[4,5].

*Akkermansia muciniphila*, one of the most abundant members of the gut microbiota, is a promising next-generation probiotic agent owing to its role in regulating host health[6,7]. Accumulating evidence

supports its beneficial effects in the maintenance of gut health and amelioration of inflammation, as well as improvement of obesity, glucose homeostasis, and associated complications[8–10]. The positive effects of *A. muciniphila* are mediated through the interaction of its bioactive components, such as outer membrane (OM) protein Amuc_1100, secreted proteins P9 and threonyl-tRNA synthetase (*Am*TARS), and EV, with the host[11–14]. *A. muciniphila*-derived bioactive molecules with potential capacity to target signaling pathways have only been recently identified; hence, further research is required to understand the interactions between the host and *A. muciniphila*.

The intestinal epithelium is one of the most rapidly self-renewing tissues with a turnover time of about 3-5 days in adult mammals[15]. The self-renewal and regenerative capacity is important for maintaining intestinal epithelium and protecting against a variety of insults from the gut lumen. This homeostatic regeneration of the intestinal epithelium relies on the active intestinal stem cells (ISCs), also called crypt-base columnar cells that reside at the base of the intestinal crypt. These cells can be identified by the expression of *Lgr5* and *Olfm4* within the murine and human small intestines[15–17]. Active ISCs divide and generate either ISC daughters or proliferating transit-amplifying (TA) progenitor cells, which terminally differentiate into secretory (enteroendocrine cells, Paneth cells, and goblet cells) and absorptive (enterocytes) cell lineages[18]. In addition, there is another stem cell pool residing at position +4 from the crypt base, and these cells are also known as reserve ISCs marked by high expression of *Msi1*, *Hopx*, and *Bmi1*[19,20]. While active ISCs are injury-sensitive, reserve ISCs are injury-resistant and contribute to injury-induced regeneration by reactivating to replenish the active ISCs pool, but are not or minimally required under normal homeostatic renewal[21].

The gastrointestinal (GI) system is particularly susceptible to exposure to a high dose of ionizing radiation and chemotherapeutic agents, resulting in the rapid depletion of active ISC populations before the occurrence of effective reserve ISCs division and crypt regeneration[22–24]. Impaired intestinal regeneration leads to the disintegration of intestinal epithelium resulting in lethal GI symptoms such as diarrhea, weight loss, and mortality, collectively known as GI syndrome[25–27]. Therefore, improvement of intestinal regeneration through regulation of ISC function is a potentially core factor in the maintenance of gut homeostasis.

In the present study, we aimed to identify new probiotic-associated effector molecules in *A. muciniphila* that play a role in modulating host cells and determine their functionality. We found that Amuc_1409 is commonly identified as a secreted protein in the cell-free supernatant of *A. muciniphila*. Subsequent in vivo and ex vivo analyses demonstrated its biological role as a beneficial effector molecule for ISC-mediated intestinal regeneration through interaction with E-cadherin, thereby activating the Wnt/β-catenin signaling pathway. Collectively, our findings suggest that *A. muciniphila*-secreted Amuc_1409 acts as a novel regulatory mediator of ISC homeostasis, providing insights into understanding the mode of action of gut microbiota in improving gut health.

## Results

### Amuc_1409 is the most commonly identified protein in the *A. muciniphila* secretome

To unearth *A. muciniphila*-derived bioactive factors, we first performed proteomic profiling of the secretome of *A. muciniphila* cultivated under basal medium using nanoflow liquid chromatography-tandem mass spectrometry (nLC-MS/MS). A total of 2216 unique peptides corresponding to 325 proteins were identified in three biological replicates (Supplementary Table 1). Of these, 285 proteins, having two or more unique peptides per protein, were considered as confident identifications (Supplementary Table 2) and were followed by an *in-silico* prediction analysis of proteins containing secretion

signal peptides using the SignalP 5.0 and SecretomeP 2.0 prediction tools. The results from these analyses showed that 32 and 46 proteins were assigned to classically and non-classically secreted proteins, respectively (Fig. 1a and Supplementary Table 3). Combined with these predicted proteins, 60 signal peptide-containing proteins were predicted to be involved in the secretome of *A. muciniphila*.

Next, we analyzed the subcellular localization of signal peptide-containing proteins using PSORTb 3.0. Proteins were classified into five groups according to their subcellular localization: cytoplasm ($n = 19$), cytoplasmic membrane ($n = 5$), periplasmic (PP) space ($n = 5$), OM ($n = 2$), and unknown (or multiple localization sites, $n = 29$) (Fig. 1b and Supplementary Table 3). Excluding proteins predicted to be located in the cytoplasm, cytoplasmic membrane, PP space, or OM, the remaining 29 proteins belonging to the unknown (or multiple localization sites) group were identified as putative extracellular proteins secreted from *A. muciniphila* (Supplementary Table 4).

Among these 29 putative extracellular proteins, PepSY_like domain-containing protein (encoded by Amuc_1409, 16.5 kDa) was the most abundant protein based on MS/MS counts and intensity-based absolute quantification (iBAQ) values (Fig. 1c). We identified eight unique peptides from Amuc_1409, accounting for 52% of the sequence coverage. The representative unique peptide of Amuc_1409 is shown in Fig. 1d.

To confirm whether Amuc_1409 is identified as an extracellular protein in *A. muciniphila* cultured under different growth conditions, we carried out a proteomic analysis of the secretome of *A. muciniphila* grown in brain-heart infusion (BHI) medium. In total, 38 unique peptides corresponding to 22 proteins were identified in BHI medium (Supplementary Table 5). According to the bioinformatics pipeline described above for predicting extracellular proteins, three proteins (encoded by Amuc_1409, Amuc_2057, and Amuc_1143) were predicted as putative extracellular proteins (Supplementary Table 6). Among these, Amuc_1409 was identified with high confidence (five unique peptides, 32.7% of sequence coverage). Taken together, Amuc_1409 was commonly identified in the secretome of *A. muciniphila* grown under different conditions: basal and BHI media. In previous proteomic studies with *A. muciniphila*, Amuc_1409 has been identified in both the cellular proteome and the supernatant fraction[11,28]; still, its biological function has not yet been elucidated. Therefore, in this study, we considered Amuc_1409 as a potential probiotic factor and investigated its biological role and mechanism of action.

### Amuc_1409 promotes ISC-mediated epithelial development in intestinal organoids by activation of the Wnt/β-catenin signaling pathway

Recent studies have reported that *A. muciniphila*, in close contact with the intestinal epithelial cells plays a critical role in ISC-mediated epithelial development and the maintenance of the gut microenvironment[29–31]. Hence, we hypothesized that Amuc_1409 may act as a beneficial effector molecule in the development and differentiation of the intestinal epithelium, due to its potential to reach and communicate with host cells. To test this hypothesis, we treated intestinal organoids generated from mouse small intestinal crypts (mIOs) with purified His-tagged Amuc_1409 (hereafter called Amuc_1409*) produced in *Escherichia coli* (*E. coli*) (Fig. 2a, b). Treatment with Amuc_1409* significantly upregulated the expression of markers for intestinal secretory cell lineages including secretory progenitors, enteroendocrine cells, Paneth cells, and goblet cells, as well as mature absorptive enterocytes (Fig. 2c–k). As mature intestinal epithelial cells terminally differentiated from ISCs[15], we analyzed the number of lobes per mIO as a quantitative indicator of the function of ISCs in Amuc_1409*-treated mIOs. Treatment with Amuc_1409*

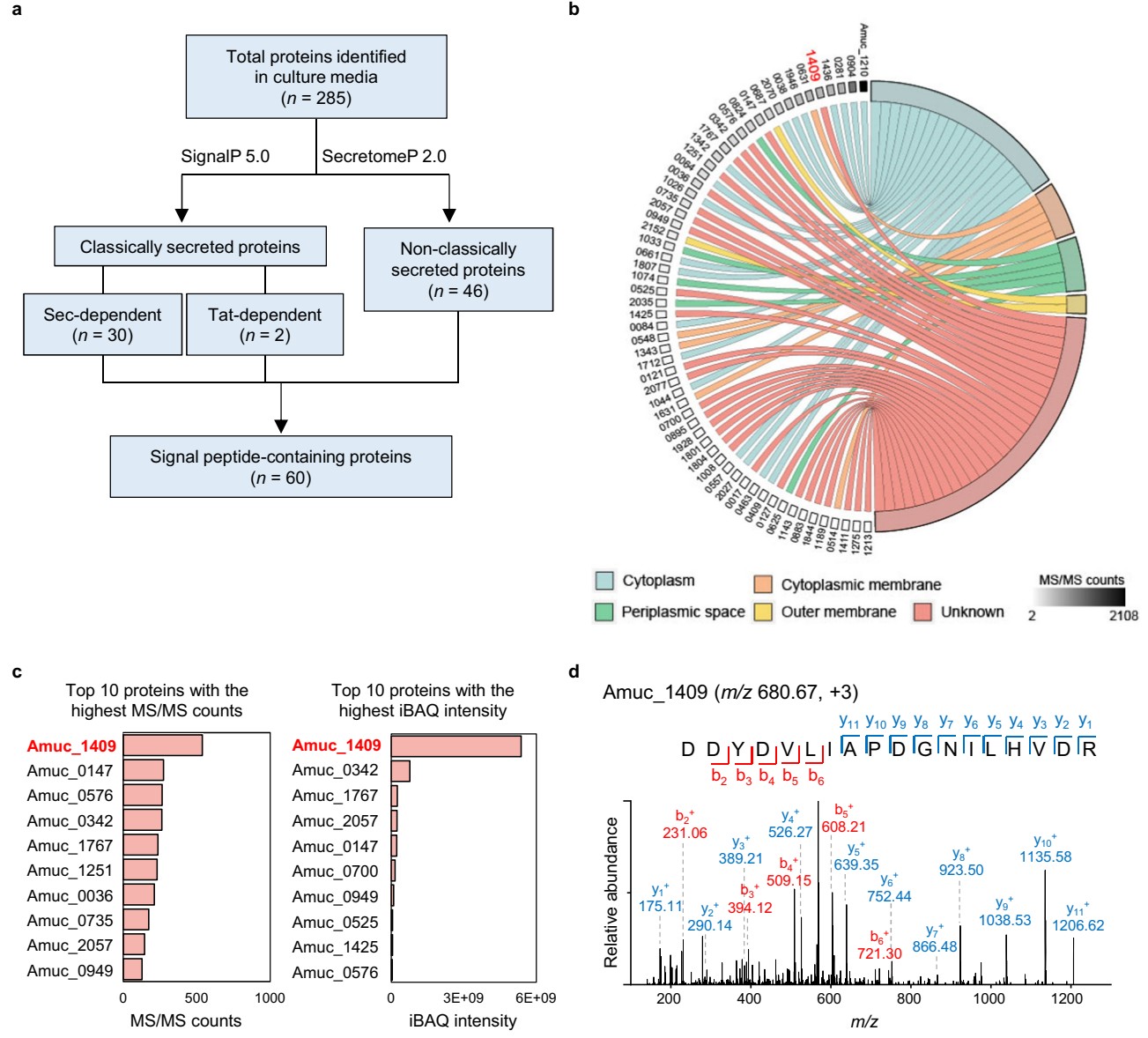

**Fig. 1 | Proteomic profiling of the A. muciniphila secretome. a** Scheme showing the workflow for the identification of signal peptide-containing proteins in the supernatant of three independently cultivated *A. muciniphila* under basal medium. **b** Chord diagram illustrating the subcellular localization of signal peptide-containing proteins as predicted with PSORTb 3.0. **c** Top 10 proteins with the highest MS/MS counts (left) and iBAQ intensity (right). **d** MS/MS spectrum of representative peptide derived from Amuc_1409. Source data are provided as a Source Data file.

promoted mIO growth (Fig. 3a) and increased the number of lobes per mIO in a dose-dependent manner (Fig. 3b and Supplementary Fig. 1a). The expression of genes related to cell proliferation and ISC was significantly upregulated in Amuc_1409*-treated mIOs compared to that in untreated mIOs (Fig. 3c). Further, Amuc_1409* enhanced Ki67 protein expression in mIOs (Fig. 3d and Supplementary Fig. 1b).

Then, we examined the effects of Amuc_1409 on ISCs using three human intestinal organoids (hIOs) derived from two human embryonic stem cell (hESC) lines and one human induced pluripotent stem cell (hiPSC) line. Here, Amuc_1409* treatment induced a significant increase in organoid size and the average number of lobes per hIO compared to that in control hIOs (Fig. 3e, f and Supplementary Fig. 1c–f). The gene expression of markers for cell proliferation, ISC, and intestinal maturation was significantly upregulated in Amuc_1409*-treated hIOs compared to that in control hIOs (Fig. 3g). Consistent with the gene expression changes, the protein expression of Ki67, achaete-scute complex homolog 2 (ASCL2), and olfactomedin 4 (OLFM4)

increased greatly in Amuc_1409*-treated hIOs (Fig. 3h and Supplementary Fig. 1g, h).

The Wnt/β-catenin signaling pathway acts as a key regulator of ISC self-renewal and differentiation during intestinal homeostasis and regeneration[32,33]. Therefore, we investigated its role in the Amuc_1409-mediated enhancement of ISC proliferation and differentiation. Treatment of mIOs with Amuc_1409* significantly upregulated the expression of Wnt/β-catenin signaling-related genes compared with that in control mIOs (Supplementary Fig. 2a). Additionally, Amuc_1409* enhanced the protein levels of active and total β-catenin in mIOs in a time-dependent manner (Supplementary Fig. 2b). Furthermore, similar to the findings in mIOs, treatment of hIOs with Amuc_1409* significantly increased the expression of Wnt/β-catenin signaling-related genes compared with that in control hIOs (Supplementary Fig. 2c). Collectively, these results suggest that Amuc_1409 promotes ISC proliferation and development of intestinal organoids, at least in part, through the activation of Wnt/β-catenin signaling.

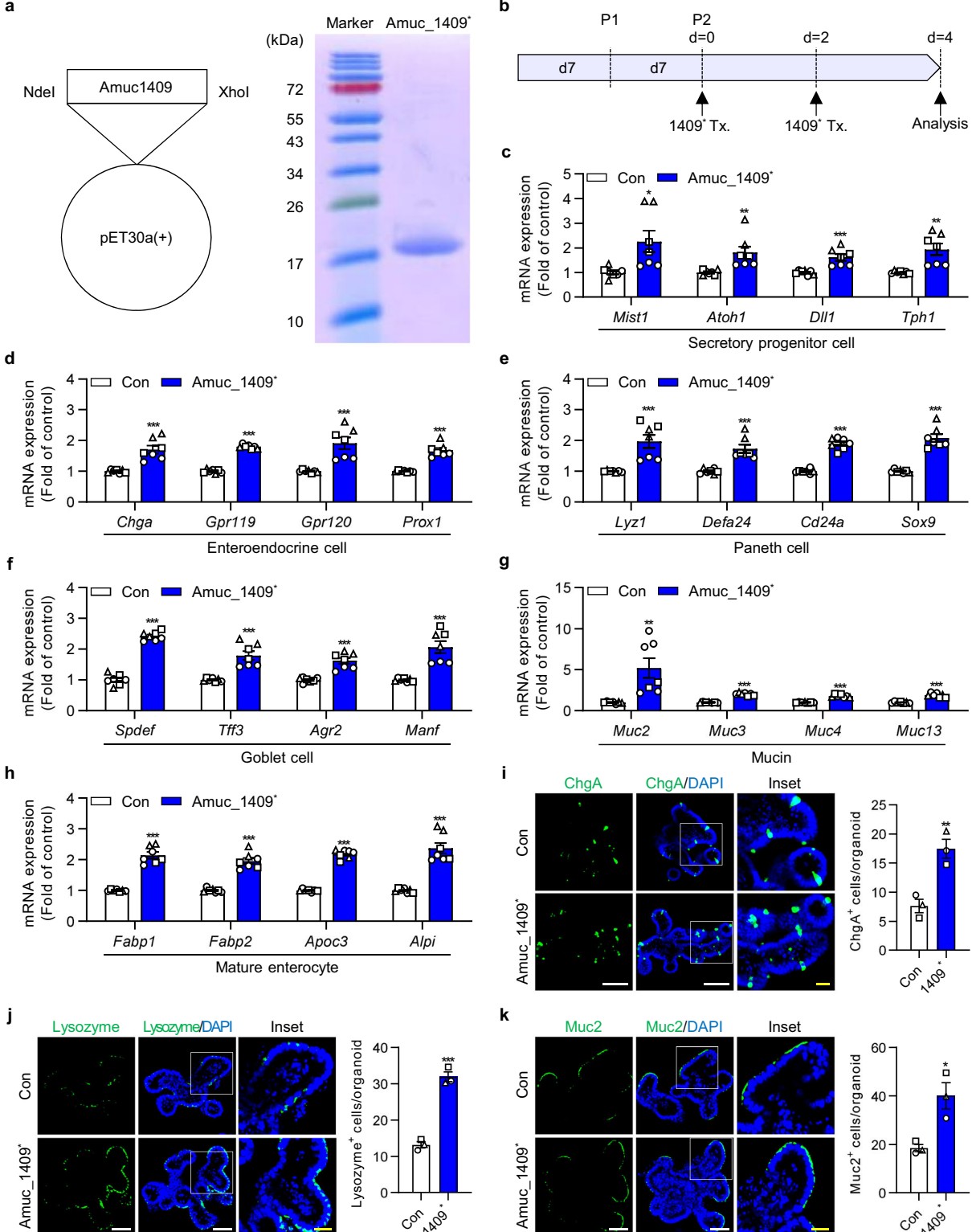

## Amuc_1409 protects from irradiation- or 5-FU-induced intestinal injury by promoting ISC-mediated regeneration

ISCs play a vital role in the maintenance of intestinal homeostasis as well as in intestinal regeneration after injury[34]. Based on our finding that Amuc_1409 regulates ISCs in mIOs and hIOs, we explored whether Amuc_1409 could promote ISC-mediated intestinal regeneration after radiation-induced injury in mice. We analyzed the survival time of mice orally administered with vehicle or Amuc_1409* after irradiation (IR) with a lethal dose of 1100 Rad[35] (Fig. 4a). Oral administration of

Amuc_1409* resulted in significantly delayed mortality throughout the observation period and prolonged the duration of survival from 14 days to 18 days after IR (Fig. 4b). The gross morphology of the intestine revealed that the length of the small intestine (SI) and colon was slightly longer, albeit not statistically significant, in Amuc_1409*-treated mice than in vehicle-treated mice after IR (Supplementary Fig. 3a–d). We further observed histopathological changes in hematoxylin and eosin (H&E)-stained SI sections. Oral administration of Amuc_1409* protected mice from IR-induced histological injuries in

**Fig. 2 | Changes in expression levels of differentiated intestinal epithelial cell-associated markers in mIOs after treatment with Amuc_1409. a** His-tagged Amuc_1409 was cloned into the pET30a(+) vector using restriction endonuclease sites (NdeI/XhoI) (left panel) and the purified His-tagged Amuc_1409 protein (Amuc_1409*) was stained with Coomassie brilliant blue on a 10% SDS-PAGE gel under reducing conditions (right panel). Lane 1 shows the molecular size of standards with their apparent molecular weights (marker) in kDa. Lane 2 shows Amuc_1409*. **b** Schematic diagram of Amuc_1409* treatment (Tx.) schedule in young mIOs. mIOs from SI crypt of young (3-month-old) mice were treated with Amuc_1409* (8 nM) on day 0 and day 2 after the second subculturing passage and harvested on day 4. qRT-PCR results showing the relative mRNA expression of secretory progenitor cell markers (**c**), enteroendocrine cell markers (**d**), Paneth cell markers (**e**), goblet cell markers (**f**), mucin markers (**g**), and mature enterocyte markers (**h**) in young mIOs treated with or without Amuc_1409* (8 nM).

Representative IF staining (left panel) for the enteroendocrine cell marker (ChgA) (**i**), Paneth cell marker (Lysozyme) (**j**), and goblet cell marker (Muc2) (**k**) in young mIOs treated with or without Amuc_1409* (8 nM). ChgA, Lysozyme, and Muc2 (green); DAPI (nuclei, blue). White scale bar, 50 μm. Yellow scale bar, 20 μm. Quantification data of positive cells for each marker per organoid are presented (**i**–**k**, right panel). All data are presented as the mean ± SEM. In (**c**–**k**), a different symbol indicates a data point representing each biological replicate from independently established organoid lines derived from distinct mouse litters (*n* = 3 biologically independent mice). In (**c**–**h**), each biological replicate includes two or three technical replicates. In (**i**–**k**), positive cells for each marker were counted in 20 organoids per group from each biological replicate. Statistical analyses were performed using two-tailed Student's *t* test (**c**–**k**) (*$p < 0.05$, **$p < 0.01$, and ***$p < 0.001$ vs control group). Source data, including the exact *p* values, are provided as a Source Data file.

the SI, such as destruction of villi and crypts and denudation of the surface epithelium (Fig. 4d, left panel), and significantly reduced histological scores (Fig. 4d, right panel). Moreover, there was a significant increase in the percentage of bromodeoxyuridine (BrdU)-positive cells in the crypts of Amuc_1409*-treated mice after IR compared with that in the vehicle group (Fig. 4e). Consistently, the gene expression of cell proliferation markers was significantly upregulated in the SI of Amuc_1409*-treated mice (Fig. 4f), suggesting that Amuc_1409 drives intestinal regeneration through elevated compensatory epithelial proliferation within crypts after IR. We assessed the ability of Amuc_1409 to promote ISC renewal in SI after IR. The gene expression of ISC markers was significantly upregulated in Amuc_1409*-treated SI mice after IR (Fig. 4g), which was further confirmed by a significant increase in OLFM4-positive cells per crypt in Amuc_1409*-treated mice using immunohistochemical (IHC) staining (Fig. 4h).

We also confirmed the protective role of Amuc_1409 in a chemotherapeutic drug (5-fluorouracil, 5-FU)-induced gut injury model (Fig. 5a). Amuc_1409* administration significantly improved 5-FU-induced body weight loss, diarrhea, and histopathological changes in SI tissues (Fig. 5b–d) with a slight increase in length of the SI and colon compared with those of vehicle-treated mice (Supplementary Fig. 3e–h).

Furthermore, similarly to the IR results, ISC renewal and compensatory epithelial proliferation effects were observed in the SI of Amuc_1409*-treated mice at day 6 post-5-FU treatment (Fig. 5e–h). Additionally, Amuc_1409* administration significantly upregulated the expression of Wnt/β-catenin signaling-related genes in SI following both IR and 5-FU treatment (Figs. 4i, 5i), suggesting that Amuc_1409 protects against radiation- or 5-FU-induced intestinal injury by promoting compensatory epithelial proliferation and ISC regeneration via activation of the Wnt/β-catenin signaling pathway.

## Amuc_1409 restores number and function of ISCs during aging

Aging has been known to cause ISC function decline, which is the main reason for impaired regenerative capacity of the intestine upon aging[36,37]. Therefore, we examined whether Amuc_1409 affected the aging-associated reduced function of ISCs. First, we examined the effect of aging on ISCs and proliferating cells in the SI crypts of young (3–4-month-old) and aged (25-month-old) mice. Consistent with the results of recent studies[38,39], aged mice showed significantly reduced OLFM4-positive ISCs (Supplementary Fig. 4a) and Ki67-positive proliferating cells (Supplementary Fig. 4b) in SI crypts compared with those in young mice. Next, we compared the efficiency of de novo crypt formation in organoids derived from SI crypts of young and aged mice to determine the regenerative capacity of ISCs. Consistent with previous findings[40,41], organoids derived from aged intestinal crypts showed smaller and fewer crypt-like structures than those derived from young intestinal crypts, suggesting reduced proliferative and regenerative capacities of aged ISCs (Supplementary Fig. 4c–f).

To examine the effect of Amuc_1409 on aged ISCs in vivo, we administered Amuc_1409* to aged mice via oral gavage for 15 weeks (Fig. 6a). There was a significant increase in the expression of markers for ISC and cell proliferation in the SI of Amuc_1409*-treated aged mice (Fig. 6b, c). Moreover, IHC staining results revealed a significantly larger number of OLFM4- and Ki67-positive cells per crypt in Amuc_1409*-treated aged mice than in untreated mice (Fig. 6d, e). We further confirmed the effect of Amuc_1409 on aged ISCs using mIOs derived from the SI crypts of aged mice (Fig. 6f). We observed that Amuc_1409* treatment increased the number of lobes per mIO (Fig. 6i, j) as well as the budding rate (Fig. 6g, h) in a dose-dependent manner. Consistent with the in vivo findings, the gene expression of markers for ISC and cell proliferation was significantly upregulated in Amuc_1409*-treated aged mIOs (Fig. 6k, l). These results suggest that Amuc_1409 can restore the number and proliferative function of ISCs in mice during aging. Reportedly, downregulation of Wnt signaling is one of the main causes of the functional decline of ISCs, and Wnt supplementation can restore the regenerative capacity of aged ISCs[36,40]. Herein, Amuc_1409* treatment increased the protein levels of active and total β-catenin in aged mIOs (Fig. 6m) and significantly increased the expression of Wnt/β-catenin signaling-related genes in aged mIOs and the SI of aged mice (Fig. 6n, o). These results suggest that Amuc_1409 can reverse the aging-induced decline in number and function of ISCs by enhancing Wnt/β-catenin signaling.

## Amuc_1409 promoted E-cadherin/β-catenin complex dissociation through an interaction with E-cadherin, thereby activating Wnt/β-catenin signaling

Then, we investigated the underlying mechanism by which Amuc_1409 activates Wnt/β-catenin signaling to regulate ISC proliferation and regeneration. By performing an in vitro binding assay to identify the binding receptor for Amuc_1409 among the typical membrane-associated molecules involved in the β-catenin-dependent Wnt pathway[42], we found that Amuc_1409* interacts with endogenous E-cadherin but not with Frizzled-7, LDL receptor related protein (LRP)5 or LRP6 (Fig. 7a). Furthermore, we observed that Amuc_1409* and E-cadherin were co-localized in the plasma membrane of intact cells, providing additional evidence for their interaction (Supplementary Fig. 5a). We used His-tagged P9 (secreted protein) and His-tagged Amuc_1100 (OM protein), well-known proteins derived from *A. muciniphila*, as control proteins and they did not bind to E-cadherin, suggesting a specific interaction between Amuc_1409 and E-cadherin (Supplementary Fig. 5b). To further identify the domain within E-cadherin that interacts with Amuc_1409, we constructed Strep-tagged full-length (Full), extracellular (EC), and intracellular (IC) E-cadherin expression plasmids (Fig. 7b, upper panel). Results from a Strep pull-down assay of each construct plus E-cadherin revealed that Amuc_1409* bound directly to the EC domain of E-cadherin and to Full E-cadherin, but not to the IC domain (Fig. 7b, lower panel).

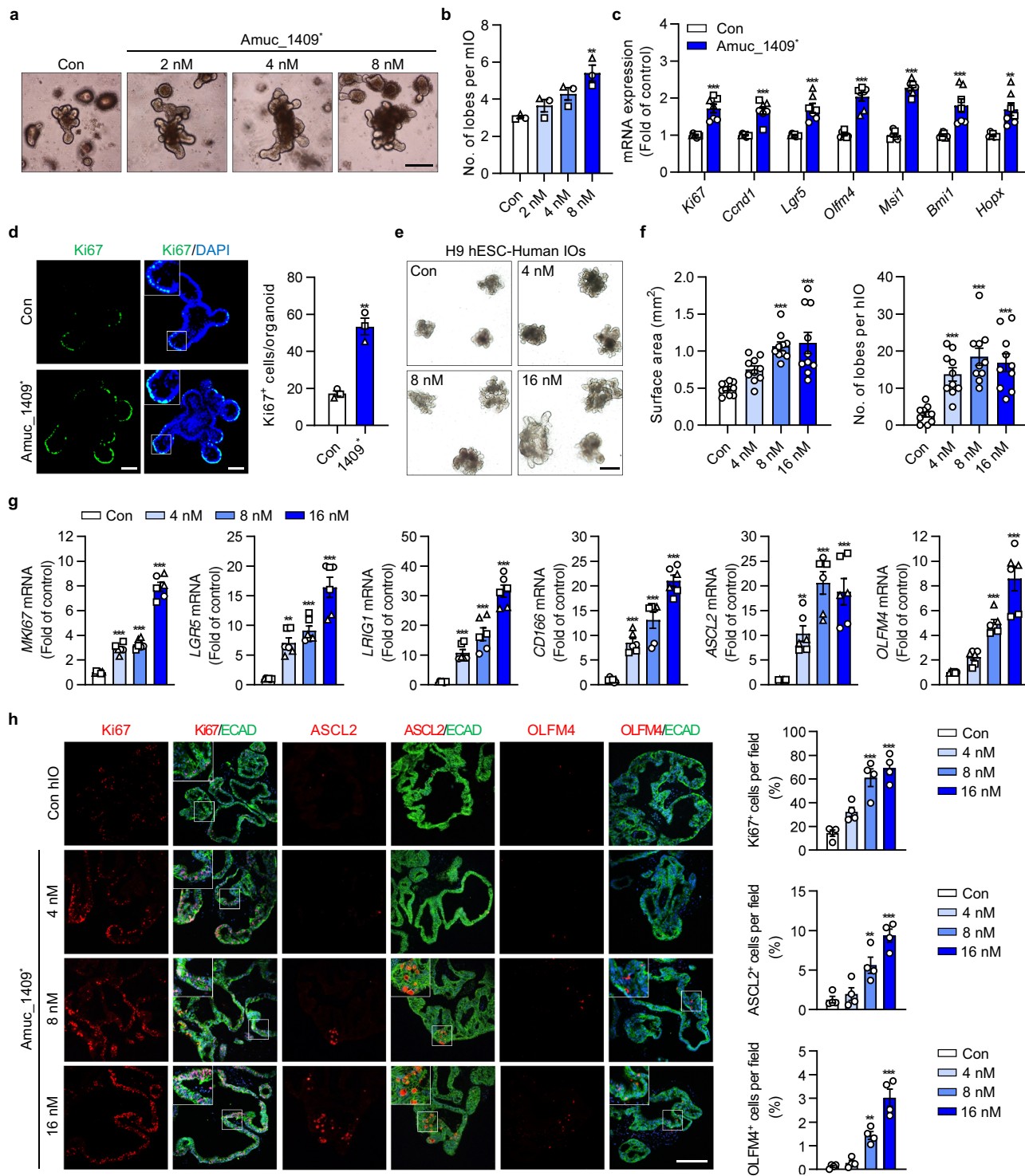

Next, we examined the signaling pathway downstream to the binding of Amuc_1409 to E-cadherin. Interaction of E-cadherin on ISCs with integrin αEβ7 on T cells reportedly triggers endocytosis of E-cadherin and the subsequent release of β-catenin from E-cadherin, which regulates ISC differentiation by activating the Wnt/β-catenin pathway[43]. Herein, cell fractionation results showed a decrease in the membrane fraction of E-cadherin and β-catenin, whereas the expression of E-cadherin and β-catenin was increased in the cytosolic fraction in response to Amuc_1409* treatment (Fig. 7c, d). These findings were additionally confirmed with confocal microscopy, whose results showed that Amuc_1409* stimulated the internalization of E-cadherin and β-catenin from the membrane to the cytosol with subsequently increased

translocation of β-catenin into the nucleus (Fig. 7e). Further validation of this Amuc_1409*-induced internalization of E-cadherin in GFP-expressing Lgr5+ ISCs was conducted using mIOs derived from *Lgr5-EGFP-IRES-CreERT2* mice (Supplementary Fig. 5c). Additionally, the results from a co-immunoprecipitation assay revealed that treatment with Amuc_1409* disrupted the E-cadherin/β-catenin interaction complex in HT-29 cells, regardless of the absence or presence of Wnt3a (Supplementary Fig. 5d). The effect of Amuc_1409* on β-catenin activity was further confirmed by the observed increase in protein levels of active and total β-catenin (Fig. 7f) and significant upregulation of β-catenin target genes (Fig. 7g) in Amuc_1409*-treated mIOs. Taken together, these results suggest that Amuc_1409 enhances Wnt/β-catenin

**Fig. 3 | Amuc_1409 promotes the proliferation and development of intestinal organoids.** Representative brightfield images (**a**) and the number of lobes per mIO (**b**) in control and Amuc_1409*-treated young mIOs. Scale bar, 200 μm. **c** Relative mRNA expression of cell proliferation (*Ki67* and *Ccnd1*) and ISC (*Lgr5, Olfm4, Msi1, Bmi1,* and *Hopx*) markers in control and Amuc_1409*(8 nM)-treated young mIOs. **d** Representative IF staining images (left panel) for Ki67 (green) and DAPI (nuclei, blue), and quantification of Ki67⁺ cells per organoid (right panel) in control and Amuc_1409*(8 nM)-treated young mIOs. Scale bar, 50 μm. Representative brightfield images (**e**), the surface area of hIOs (**f**, left panel), and the number of lobes per hIO (**f**, right panel) in control and Amuc_1409*-treated hIOs derived from H9 hESC line. *n* = 10 organoids per group. Scale bar, 500 μm. **g** Relative mRNA expression of proliferation (*MKI67*), ISC (*LGR5, LRIG1, CD166,* and *ASCL2*) and intestinal maturation (*OLFM4*) markers in hIOs. **h** Representative IF staining images (left panel) for Ki67, ASCL2, and OLFM4 (red) in hIOs derived from H9 hESC line. Intestinal epithelial cells were identified using E-cadherin (ECAD, green); DAPI (blue). Scale bar, 200 μm. Quantification of the percentage of positive cells for each marker per field of view is presented (right panel, *n* = 4 fields per group). All data are presented as the mean ± SEM. In (**b**–**d**) and (**g**), a different symbol indicates a data point representing each biological replicate from independently established organoid lines derived from distinct mouse litters (**b**–**d**: *n* = 3 biologically independent mice) or two hESC/one hiPSC lines (**g**: *n* = 3 biologically independent organoid lines). In (**c**) and (**g**), each biological replicate includes two or three technical replicates. In (**d**), Ki67⁺ cells were counted in 20 organoids per group from each biological replicate. Statistical analyses were performed via one-way ANOVA with Dunnett's multiple comparisons test (**b, f, g, h**) and two-tailed Student's *t* test (**c, d**) (*$p < 0.05$, **$p < 0.01$, and ***$p < 0.001$ vs control group). Source data, including the exact *p* values, are provided as a Source Data file.

signaling through inducing dissociation of E-cadherin/β-catenin complex following an interaction with EC-domain of E-cadherin.

## Amuc_1409 directly promotes the regenerative function of ISCs in an E-cadherin-dependent manner

Finally, we examined whether E-cadherin-dependent signaling is required for the Amuc_1409-mediated enhancement of ISC function using intestinal crypt organoids from *Lgr5-CreERT2;Cdh1ᶠˡ/ᶠˡ* mice, in which Cre recombinase was inducible with 4-hydroxytamoxifen (4-OHT) (Fig. 8c). *Cdh1* gene expression decreased following 4-OHT treatment (Fig. 8a) and the specific deletion of E-cadherin in GFP-expressing Lgr5⁺ ISC was confirmed at the protein level (Fig. 8b, white arrowheads). As expected, the deletion of E-cadherin in ISCs abrogated the effects of Amuc_1409* on mIO growth (Fig. 8d–f). Moreover, the Amuc_1409*-mediated induction of genes related to ISC, cell proliferation, differentiated intestinal epithelial cell, and β-catenin signaling was discarded in ISC-specific E-cadherin lacking mIOs (Fig. 8g, h and Supplementary Fig. 6a–f). Collectively, our data suggest that interaction with E-cadherin is indispensable for Amuc_1409-mediated improvement in the regenerative capacity of ISCs.

## Discussion

*A. muciniphila* has been known as a key component of the gut microbiome with many beneficial biological functions. However, bioactive molecules with potential signaling capacity derived from *A. muciniphila* are only starting to be discovered. Further research, therefore, is needed for a better understanding of the interaction between the host and microbiome. Herein, we investigated the composition of extracellular proteins secreted by *A. muciniphila* cultured with basal and BHI medium using an MS-based proteomic analysis. We determined that Amuc_1409, a previously uncharacterized protein, was a commonly identified protein in the secretome of *A. muciniphila* grown under different conditions and, for the first time, characterized its beneficial effect on ISC-mediated intestinal regeneration. We found that Amuc_1409 improves intestinal homeostasis by enhancing the regenerative capacity of ISCs during radiation- or chemotherapeutic drug-induced intestinal injury and aging (Fig. 8i, upper panel). The potential role of Amuc_1409 in ISC regeneration is mediated by the promotion of E-cadherin internalization and the subsequent dissociation of β-catenin from E-cadherin through its interaction with the EC domain of E-cadherin, resulting in the activation of Wnt/β-catenin signaling (Fig. 8i, lower panel).

The gut microbiota is restricted from directly interacting with the intestinal epithelium because it is covered by a mucus layer enriched with immune effectors and antimicrobial peptides targeting the microbiota, thus providing a physical and biochemical barrier[44,45]. Therefore, research invested in the mode of action of gut microbiota on host health has acquired more interest in microbiota-secreted factors such as protein, metabolites, and EV that have the potential to penetrate the intestinal mucus layer and reach host cells, as they are interesting candidates for mediating host-microbiome crosstalk[4]. Recently, P9 was identified in the cell-free supernatant of *A. muciniphila* culture as a new promising probiotic effector that promotes the ability of the bacterium to improve metabolic syndrome in mice with diet-induced obesity by inducing the secretion of glucagon-like peptide-1[11]. Also, secreted *Am*TARS was reported as an anti-inflammatory immune mediator by restoring IL-10-secreting macrophages through the interaction of toll-like receptor 2[14]. Our findings support the concept that extracellular proteins secreted by probiotic bacteria can mediate beneficial effects on host health by functioning as arbitrators for host-microbiome interactions.

*A. muciniphila* is a gram-negative bacterium enveloped with two membranes, the inner and the outer membrane, that are separated by the periplasm[46], and there is still a possibility that Amuc_1409 may be in a different compartment within the cell, albeit it contains predicted signal sequences. Given this consideration, we further explored whether Amuc_1409 is exclusively present as a secreted protein or could also localize in other subcellular space. When analyzed using western blot and mass spectrometry-based proteomic approach in the four protein-enriched fractions of *A. muciniphila* cultured under basal medium, Amuc_1409 was found to localize in WC, PP, and OM, but in homeopathic amounts compared to the abundance of previously known *A. muciniphila*-derived protein localizing to each compartment. The expression levels of Amuc_1409 in cell-free culture supernatant were measured to be highly abundant, compared to those in other subcellular locations (Supplementary Fig. 7a–c). In general, gram-negative bacteria have evolved two fundamentally different types of protein secretion systems: one-step secretion directly from the cytoplasm to the extracellular space and two-step secretion involving transport across the inner membrane to the PP by the SEC or TAT machinery, which recognizes signal sequence, followed by entry or crossing of the OM[47]. Based on this information, our observation suggested that Amuc_1409 is a secreted protein that can localize in the PP and OM during export to the extracellular space. Further investigation is required to identify the mechanism by which Amuc_1409 is secreted from the PP across the OM and the possibility of its entry into the OM from the PP space to form membrane proteins or cell-wall components, and research on the latter concept could help explore the potential of Amuc_1409 in contributing to the beneficial effects of pasteurized *A. muciniphila* in both humans and mice, considering our heat stability test that demonstrated high thermal stability of Amuc_1409 (Supplementary Fig. 8a, b).

Augmentation of the Wnt signaling pathway is required for intestinal repair after injury and improvement of the impaired regenerative capacity of aged ISCs[40,48]. β-catenin is the central downstream effector of the Wnt signaling pathway and is sequestered in the cytoplasmic domain of E-cadherin at the plasma membrane, preventing its participation in Wnt/β-catenin signaling[49]. Here, we found that Amuc_1409 regulates the regenerative function of ISCs by upregulating β-catenin activity through E-cadherin internalization. Our findings

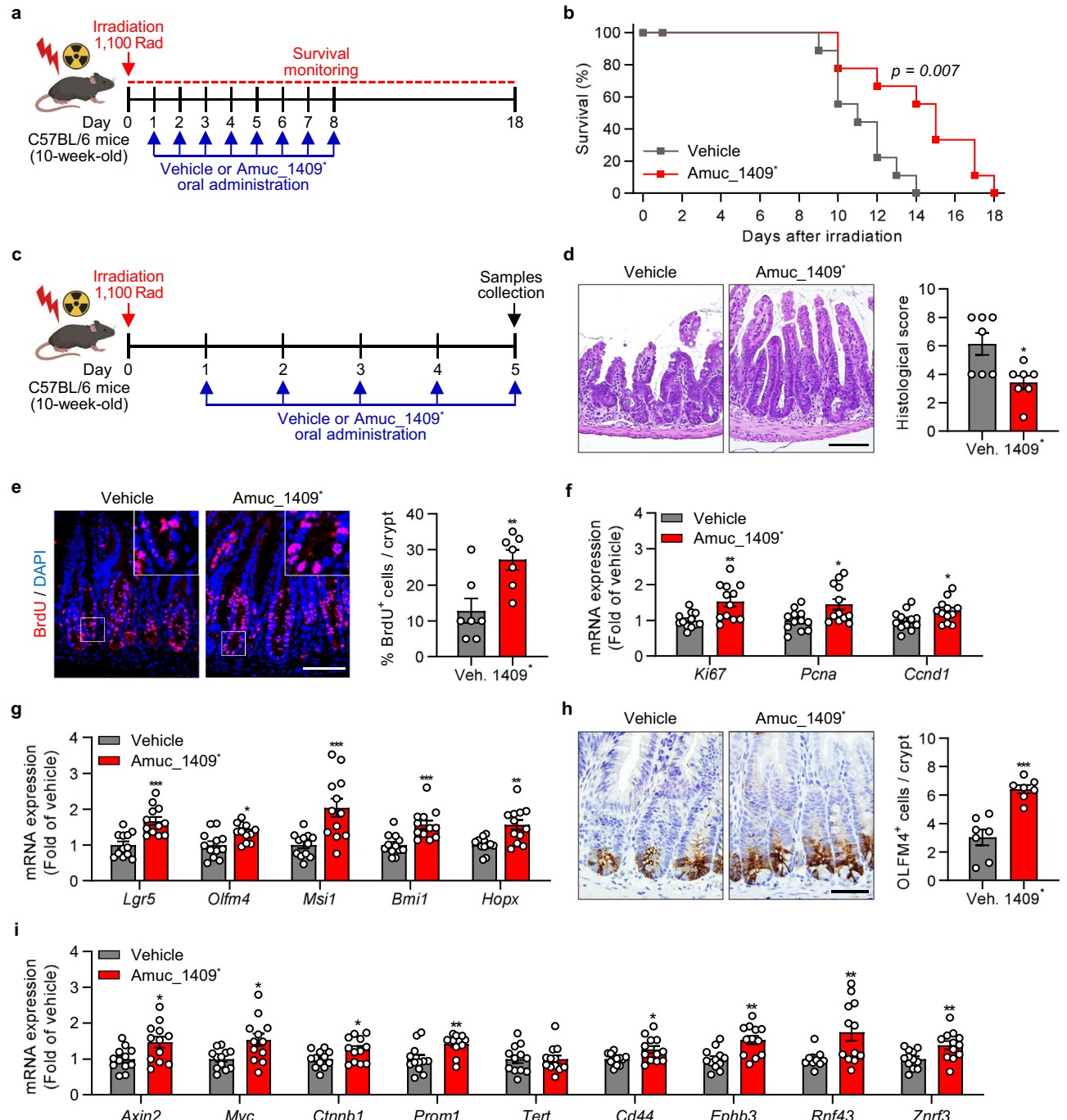

**Fig. 4 | Oral administration of Amuc_1409 protects against irradiation-induced gut damage by accelerating ISC regeneration. a** Scheme showing the experimental timeline for vehicle or Amuc_1409* (9 μg per mouse) treatment for survival rate analysis in the radiation (1100 Rad)-induced intestinal injury model. **b** The survival rate of vehicle- or Amuc_1409*-treated mice following 1100 Rad of radiation (*n* = 9 biological replicate mice per group). **c** Scheme showing the experimental workflow for oral administration of vehicle or Amuc_1409* (9 μg per mouse) for histological (**d, e, h**: *n* = 7 biological replicate mice per group) and qRT-PCR (**f, g, i**: *n* = 12 biological replicate mice per group) analysis of SI tissues in the radiation (1100 Rad)-induced intestinal injury model. **d** Representative images of H&E staining (left panel) and histological scores (right panel) in sections of SI. Scale bar, 100 μm. **e** Representative IF images of BrdU staining (left panel) and quantification of percentage of BrdU⁺ cells per crypt (right panel) in sections of SI. Scale bar,

100 μm. qRT-PCR results showing the relative mRNA expression of cell proliferation markers (*Ki67*, *Pcna*, and *Ccnd1*) (**f**) and ISC markers (*Lgr5*, *Olfm4*, *Msi1*, *Bmi1*, and *Hopx*) (**g**) in SI tissues. **h** Representative images of IHC staining for OLFM4 (left panel) and quantification of OLFM4⁺ cells per crypt (right panel) in sections of SI. Scale bar, 50 μm. **i** qRT-PCR results showing the relative mRNA expression of Wnt/β-catenin target genes in SI tissues. All data are presented as the mean ± SEM. Data shown in (**b**) and (**d**–**i**) are representative of three independent experiments, each with similar results. Each data point represents a biological replicate, corresponding to one mouse. Images in (**a**) and (**c**) were created with BioRender.com and have been granted a publication license. Statistical analyses were performed using Log-Rank test (**b**) and two-tailed Student's *t* test (**d**–**i**) (*\*p* < 0.05, *\*\*p* < 0.01, and *\*\*\*p* < 0.001 vs vehicle-treated group). Source data, including the exact *p* values, are provided as a Source Data file.

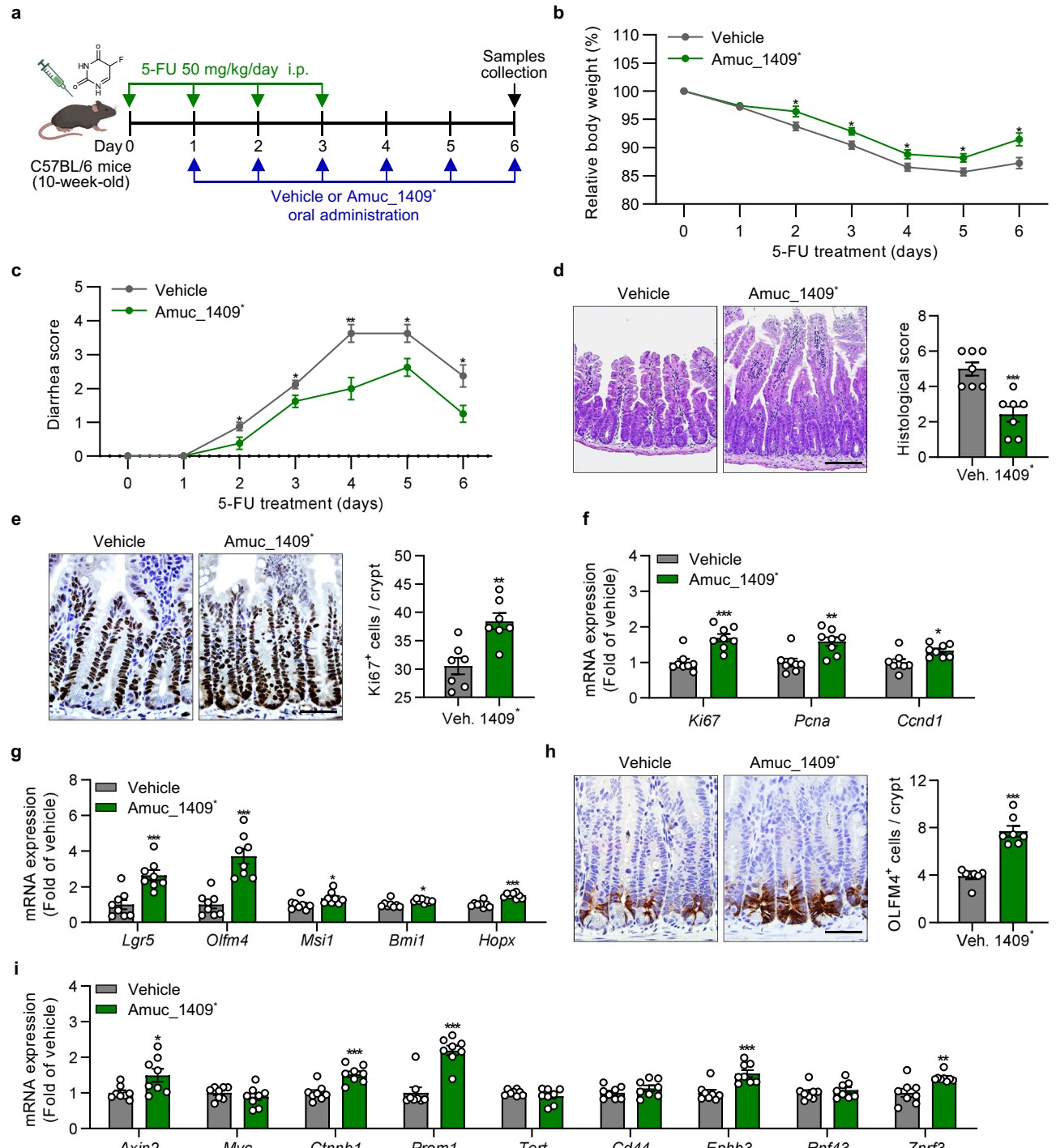

**Fig. 5 | Amuc_1409 treatment improves 5-FU-induced gut injury by promoting ISC regeneration. a** Scheme showing the experimental workflow for oral administration of vehicle or Amuc_1409* (9 μg per mouse) in the mice injected intraperitoneally (i.p.) with 5-FU (50 mg/kg/day). Assessment of the effect of Amuc_1409* on 5-FU-induced changes in body weight (**b**) and diarrhea (**c**) in mice. Body weight changes are expressed as a percentage of body weight relative to that on the day of the first 5-FU injection (day 0). **d** Representative images of H&E staining (left panel) and histological scores (right panel) in sections of SI. Scale bar, 100 μm. **e** Representative images of IHC staining for Ki67 (left panel) and quantification of Ki67+ cells per crypt (right panel) in sections of SI tissues. Scale bar, 50 μm. **f** qRT-PCR results showing the relative mRNA expression of cell proliferation markers (*Ki67, Pcna,* and *Ccnd1*) in SI tissues. **g** qRT-PCR analysis of the relative mRNA expression of ISC markers (*Lgr5, Olfm4, Msi1, Bmi1,* and *Hopx*) in SI tissues.

**h** Representative images of IHC staining for OLFM4 (left panel) and quantification of OLFM4+ cells per crypt (right panel) in SI sections. Scale bar, 50 μm. **i** qRT-PCR results showing the relative mRNA expression of Wnt/β-catenin target genes in SI tissues. All data are presented as the mean ± SEM. Data shown in (**b**–**i**) are representative of three independent experiments, each with similar results. Each data point represents a biological replicate, corresponding to one mouse (**b, c, f, g, i:** n = 8 biological replicate mice per group, **d, e, h:** n = 7 biological replicate mice per group). Images in (**a**) were created with BioRender.com and have been granted a publication license. The chemical structure in (**a**) was created with ChemDraw. Statistical analyses were performed using two-tailed Student's *t* test (**b**–**i**) (*p < 0.05, **p < 0.01, and ***p < 0.001 vs vehicle-treated group). Source data, including the exact *p* values, are provided as a Source Data file.

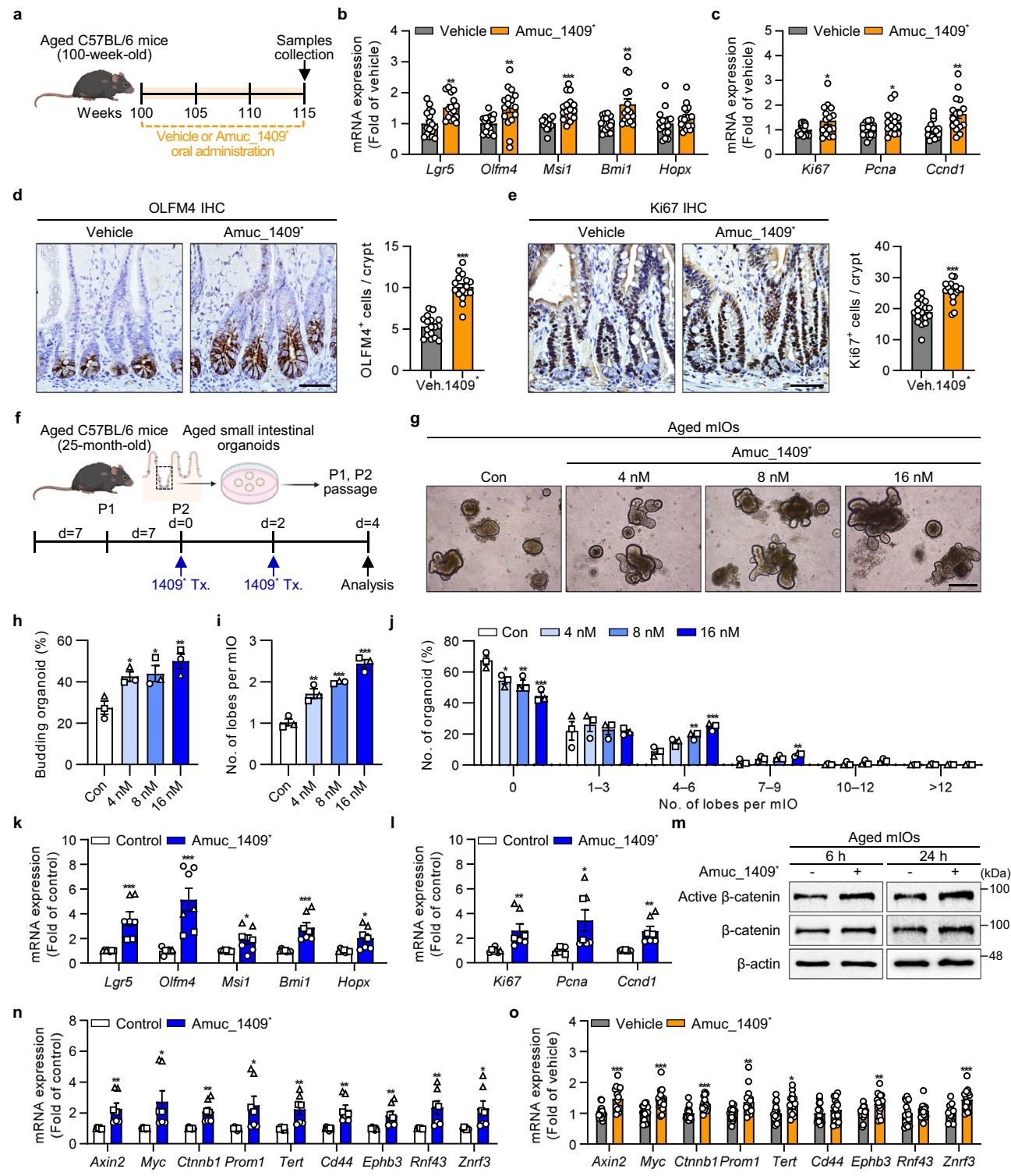

are consistent with previous studies reporting that loss of membranous E-cadherin induces the release and subsequent activation of β-catenin[43,50], and support the notion that E-cadherin plays an important role in regulating β-catenin function and stabilization. Further studies are needed to determine the mechanism whereby Amuc_1409 triggers the internalization of E-cadherin from the plasma membrane and downstream of its interaction with the EC domain of E-cadherin.

The dysregulation of the E-cadherin/β-catenin adhesion complex has reportedly been implicated in various processes involved in the development and progression of cancer, including apoptosis,

proliferation, migration, invasion, and epithelial-mesenchymal transition[51]. Therefore, we investigated the potentially undesirable effect of Amuc_1409 on tumor growth. To achieve this, we subcutaneously injected E-cadherin-expressing CT-26 cells into BALB/c mice and administered either vehicle or Amuc_1409* orally to mice for 2 weeks, starting at 1 week post-inoculation (Supplementary Fig. 9a). Within 2 weeks, there were no noticeable differences in the tumor volume and body weight between the vehicle and Amuc_1409* treatment groups (Supplementary Fig. 9b–d). Furthermore, at the end of the experiment, there was no significant difference in tumor weight

**Fig. 6 | Amuc_1409 augments the aging-induced decrease in ISC number and function. a** Scheme for Amuc_1409* (4.5 μg/mouse) treatment in natural aging mice model. Relative mRNA expression of ISC (**b**) and cell proliferation (**c**) markers in aged SI. Representative IHC images for OLFM4 (**d**, left panel) and Ki67 (**e**, left panel), and quantification of OLFM4$^+$ cells (**d**, right panel) and Ki67$^+$ cells (**e**, right panel) per crypt in aged SI. Scale bar, 50 μm. **f** Scheme for Amuc_1409* treatment (Tx.) in aged mIOs. Representative brightfield images (**g**), percentage of budding organoids (**h**), number of lobes per mIO (**i**), and percentage distribution of organoids with the indicated number of lobes per mIO (**j**) in aged mIOs. Scale bar, 200 μm. Relative mRNA expression of ISC (**k**) and cell proliferation (**l**) markers in Amuc_1409* (16 nM)-treated aged mIOs. **m** Immunoblot analysis of active β-catenin and total β-catenin in aged mIOs treated with Amuc_1409* (16 nM) for indicated time points before harvest. **n** Relative expression of Wnt/β-catenin target genes in Amuc_1409* (16 nM)-treated aged mIOs. **o** Relative expression of Wnt/β-catenin target genes in aged SI. All data are presented as the mean ± SEM. In (**b**–**e**) and (**o**), each data point represents a biological replicate, corresponding to one mouse (vehicle, $n = 17$; Amuc_1409*, $n = 16$ biological replicate mice). Data are combined from two independent experiments. In (**h**–**l**) and (**n**), a different symbol indicates a data point representing each biological replicate from independently established organoid lines derived from distinct mouse litters ($n = 3$ biologically independent mice). In (**k**, **l**), and (**n**), each biological replicate includes two or three technical replicates. Images in (**a**) and (**f**) were created with BioRender.com under a granted publication license. Statistical analyses were performed using one-way ANOVA with Dunnett's multiple comparisons test (**h**, **i**, **j**) and two-tailed Student's $t$ test (**b**–**e**, **k**, **l**, **n**, **o**) (*$p < 0.05$, **$p < 0.01$, and ***$p < 0.001$ vs control group). Source data, including the exact $p$ values and uncropped western blot images, are provided as a Source Data file.

between the two groups (Supplementary Fig. 9e, f), indicating that Amuc_1409 did not promote unlimited proliferation of tumor cells in this mouse model.

We explored both the short- and long-term beneficial effects of Amuc_1409 on ISCs by employing different administration periods in several in vivo models, thereby providing a comprehensive insight into its therapeutic potential. Exposure to high-dose radiation or the chemotherapeutic drug damages DNA and selectively targets the rapidly proliferating ISCs and TA cells in the GI tract, resulting in acute GI syndrome[52]. Therefore, we chose short-term administration of Amuc_1409* in the radiation- and 5-FU-induced gut damage models to investigate the acute effect of Amuc_1409 on ISC regenerative function following injury. On the other hand, given that aging is a gradual process occurring over an extended period[53], long-term administration in a natural aging mouse model allowed us to more comprehensively evaluate the potential effects of Amuc_1409 in slowing down or mitigating age-related functional decline. Additionally, we evaluated any undesired side effects during the 15 weeks of Amuc_1409* treatment in our natural aging mouse model and observed no signs of obvious side effects in terms of changes in body weight and blood chemistry panels (Supplementary Fig. 10a–l). Although the concentration of Amuc_1409 used in animal studies was higher than our estimated physiological level of Amuc_1409 protein (approximately 33.06 ± 3.95 ng) in the whole fecal sample per mouse (Supplementary Fig. 11a–c), our findings strongly suggest that the pharmacological use of Amuc_1409 supports ISC renewal and compensatory proliferation, contributing to efficient protection in response to gut injury and against ISC functional decline during aging without promoting unlimited proliferation or causing any undesired side effects. This approach provides a pharmacological strategy that might offer greater advantages compared to utilizing the entire microorganism for therapeutic purposes. Further studies will be necessary to clarify the precise levels of Amuc_1409 produced by *A. muciniphila* in the gut or, if it exists, in circulating form, under both healthy and diseased in vivo situations. Additionally, determining whether similar effects can be observed with more physiologically relevant concentrations would contribute to a better understanding of the mode of action of *A. muciniphila* from a microbiological perspective.

Our study used N-terminal His-tagged Amuc_1409 produced in *E. coli* and purified using NI-NTA affinity chromatography (Amuc_1409*). The His-tag label might interfere with the folding or activity of the protein[54], and contaminant carryover during the recombinant protein purification process could cause potential effects. To address the concerns related to these effects, we obtained a vehicle solution (termed here empty vector vehicle) from the same purification process of empty vector-transformed *E. coli* as a control in our mIO experiments. We also generated two variants of Amuc_1409 without His-tag. The His-tag was removed in Amuc_1409* by cleavage using the tobacco etch virus protease (termed here Amuc_1409#), and for the other variant one, after the removal of the His-tag, the protein underwent an additional purification step through gel filtration chromatography (termed here Amuc_1409$). Both types of Amuc_1409 without His-tag (Amuc_1409# and Amuc_1409$) exhibited similar effects in promoting organoid differentiation compared to the His-Amuc_1409 protein (Amuc_1409*), with no impact from the PBS or empty vector vehicle (Supplementary Fig. 12a–d). Taken together, we have clarified that the observed effects of recombinant His-tagged Amuc_1409 on ISCs were specifically attributed to the Amuc_1409 protein itself, rather than the potential contaminant carryover and the presence of the His-tag.

To the best of our knowledge, this study is the first to show that the *A. muciniphila*-derived Amuc_1409 plays a role in the regenerative function of ISCs via its direct interaction with E-cadherin. This illustrates the importance of a deeper exploration of microbiota-host interactions mediated by bioactive molecules. Our findings could open up avenues for developing a new probiotic product that improves gut health by regulating intestinal homeostasis, especially during radiation- or chemotherapeutic drug-induced intestinal damage and aging.

## Methods

### Bacterial strain and growth conditions

*Akkermansia muciniphila* Muc$^T$ (=DSM 22959$^T$) was obtained from the German Collection of Microorganisms and Cell Cultures (Leibniz-Institut DSMZ-Deutsche Sammlung von Mikroorganismen und Zellkulturen GmbH, Braunschweig, Germany). *A. muciniphila* was cultivated anaerobically at 37 °C for 48 h in two different types of media: brain–heart infusion medium (BHI; 237500, BD, Hampton, NJ, USA) and a basal medium. The basal medium contained (L$^{-1}$); 16 g soy-peptone, 25 mM glucose, 0.4 g KH$_2$PO$_4$, 0.53 g Na$_2$HPO$_4$, 0.3 g NH$_4$Cl, 0.3 g NaCl, 0.1 g MgCl$_2$.6H$_2$O, 0.11 g CaCl$_2$, 4 g NaHCO3, 0.5 g cysteine, 1 mL alkaline trace element solution, 1 mL acid trace element solution, and 1 mL vitamin solution. The alkaline trace element, acid trace element, and vitamin solutions were prepared as described previously[9]. The media were supplemented with 0.2% (w/v) mucin from porcine stomach (M1778, Sigma-Aldrich, St. Louis, MO, USA), purified via ethanol precipitation[55]. All procedures for media preparation were performed under anaerobic conditions in serum bottles sealed with butyl-rubber and gas phase N$_2$.

### Purification of mucin

25 g porcine stomach mucin was stirred for 1 h at 22 °C in 0.02 M phosphate buffer (0.4 g KH$_2$PO$_4$, 0.53 g Na$_2$HPO$_4$), 0.1 M NaCl, pH 7.8. A few drops of toluene were added, and the pH was adjusted to 7–7.4 after 1 h using 2 N NaOH. The solution was then stirred for 19 h at 22 °C. The supernatant obtained by centrifugation at 10,000 × $g$ for 1 h at 4 °C was transferred to a beaker, and pre-cooled ethanol was added to achieve a final concentration of 60% (v/v). The pellet collected after centrifugation was dissolved in 0.1 M NaOH and then precipitated again with ethanol (final concentration 60% (v/v)). The pellet collected after centrifugation was dissolved in distilled water, dialyzed using

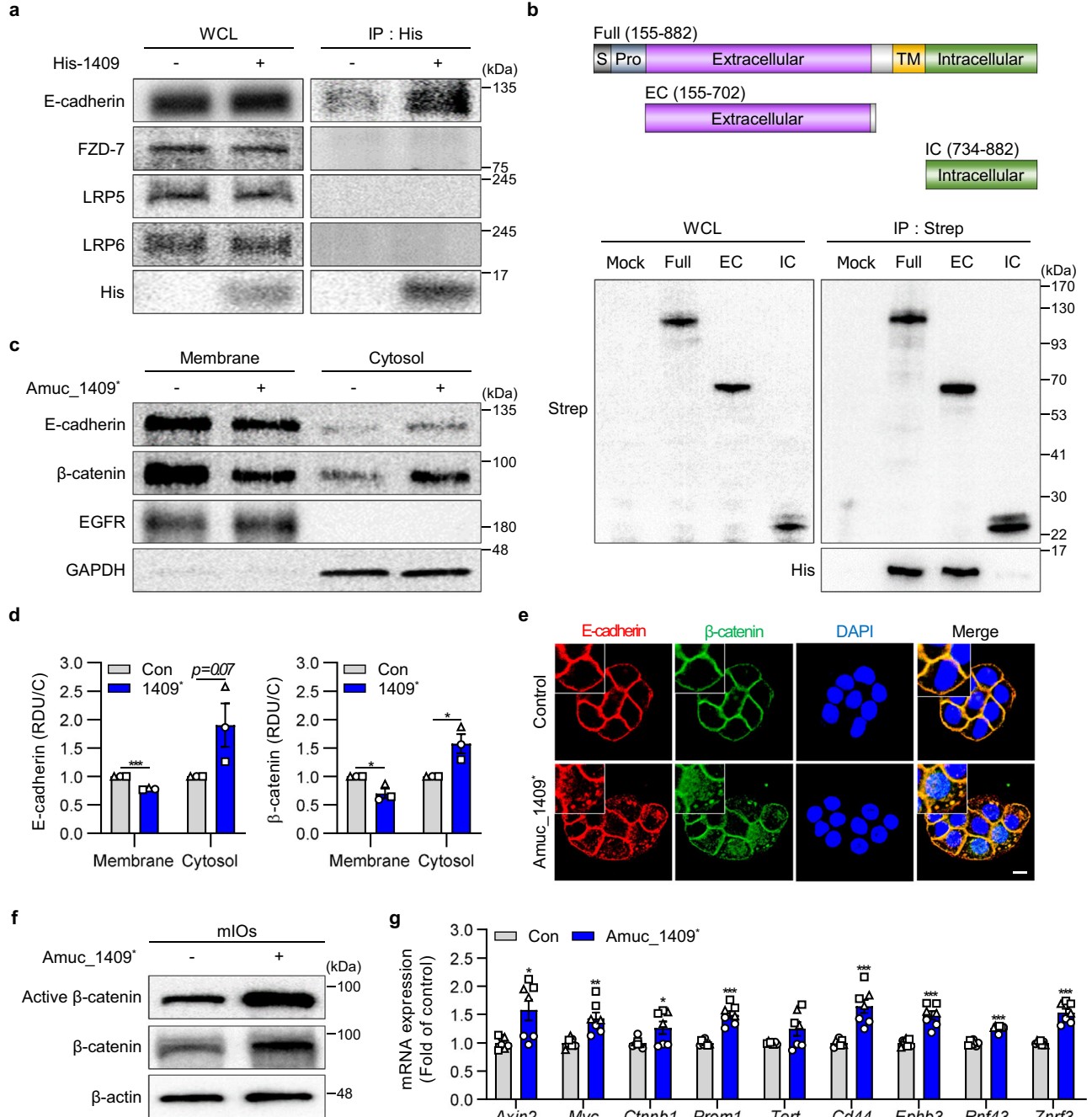

**Fig. 7 | Amuc_1409 induces dissociation of β-catenin from E-cadherin, resulting in the activation of Wnt/β-catenin signaling. a** Results after WCLs from HT-29 cells incubated with or without purified His-tagged Amuc_1409 protein (Amuc_1409*, 0.5 µg) for 1 h were subjected to immunoprecipitation using His-antibodies, followed by immunoblotting with indicated antibodies. **b** Schematics of the domain constructs of E-cadherin (upper panel); HEK293T cells were transfected with Strep-tagged E-cadherin domain constructs, and WCLs were subjected to a Strep pull-down assay. Immunoprecipitates were incubated with Amuc_1409* (20 µg), followed by immunoblotting with indicated antibodies (lower panel). Data are representative of three independent experiments. Representative immunoblot images (**c**) and relative quantitative analysis for E-cadherin (**d**, left panel) and β-catenin (**d**, right panel) of the membrane and cytosolic fractions from HT-29 cells treated with Amuc_1409* (16 nM) for 30 min. The quantification data are expressed as relative densitometer units with respect to the control group of each fraction (RDU/C). Epidermal growth factor receptor (EGFR) and GAPDH were used as

loading controls for the membrane and cytosol, respectively. The blots shown are representative of three independent experiments. **e** Representative images of IF staining for E-cadherin and β-catenin in HT-29 cells treated with Amuc_1409* (16 nM) for 30 min. E-cadherin (red); β-catenin (green), DAPI (nuclei, blue). Scale bar, 10 µm. Immunoblot analysis of active and total β-catenin (**f**) and relative mRNA expression of Wnt/β-catenin target genes (**g**) in young mIOs treated with Amuc_1409* (16 nM) for 30 min before harvest. In (**g**), a different symbol indicates a data point representing each biological replicate from independently established organoid lines derived from distinct mouse litters (n = 3 biologically independent mice). Each biological replicate includes two or three technical replicates. All data are presented as the mean ± SEM. Data shown are representative of two independent experiments, each with similar results unless otherwise stated. Statistical analyses were performed using two-tailed Student's *t* test (**d, g**) (*p < 0.05, **p < 0.01, and ***p < 0.001 vs control group). Source data, including the exact *p* values and uncropped western blot images, are provided as a Source Data file.

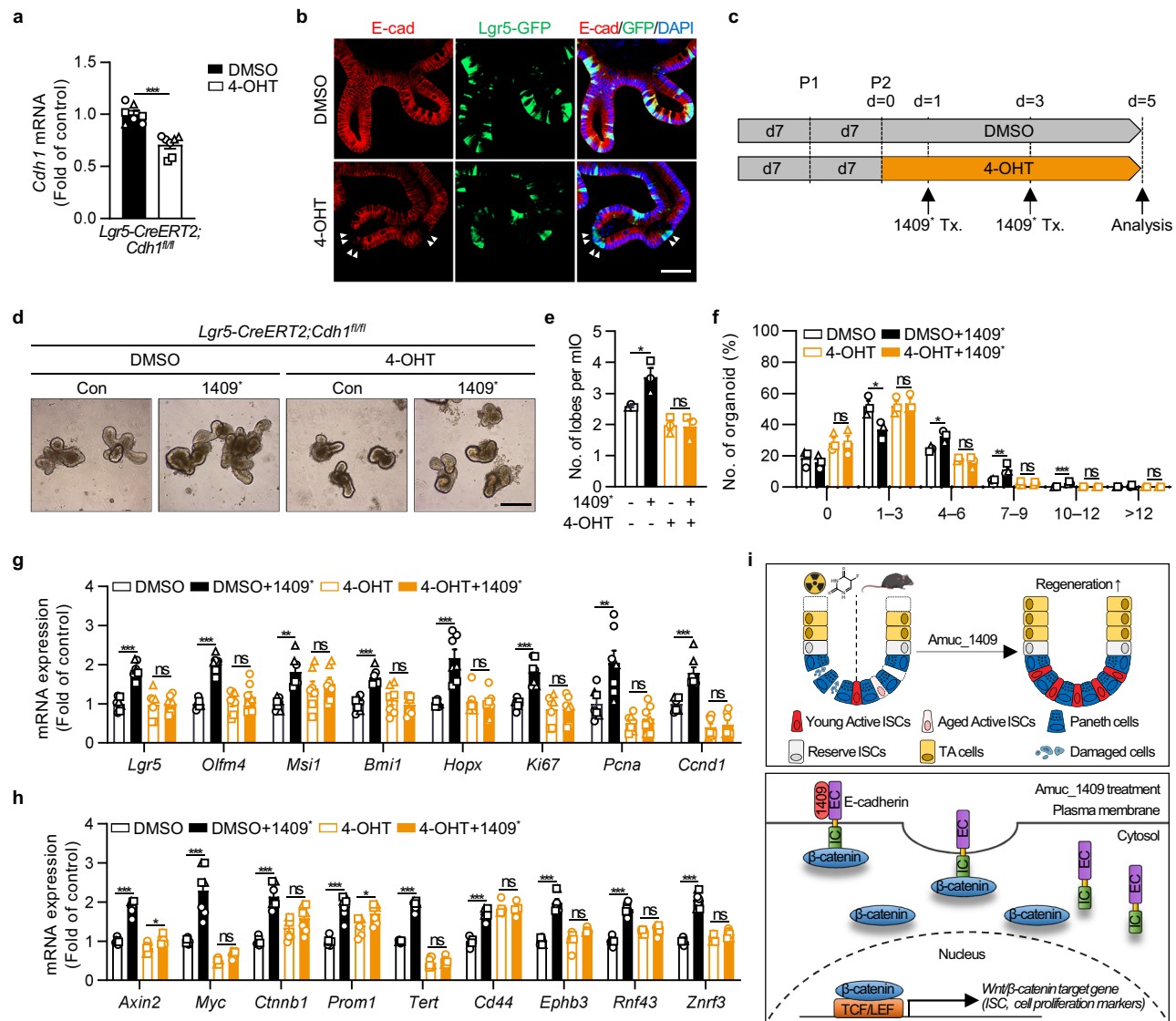

**Fig. 8 | Amuc_1409 enhances the regenerative function of ISCs in an E-cadherin-dependent manner.** qRT-PCR results showing the relative *Cdh1* (E-cadherin gene) mRNA expression (**a**) and representative IF staining for E-cadherin and GFP (**b**) to confirm the *Cdh1* deletion in GFP-expressing *Lgr5*⁺ ISCs in mIOs from *Lgr5-CreERT2;Cdh1^{fl/fl}* mice at day 5 after 4-OHT addition began. The specific deletion of E-cadherin in GFP-expressing Lgr5⁺ ISC is indicated by the white arrowheads. Scale bar, 50 µm. **c** Diagram showing the 4-OHT (1 µM) and Amuc_1409* (16 nM) treatment (Tx.) schedule in mIOs from the SI crypt of *Lgr5-CreERT2;Cdh1^{fl/fl}* mice. mIOs were analyzed on day 5 after either DMSO (vehicle control) or 4-OHT addition began. Representative brightfield images (**d**), assessment of the number of lobes per mIO (**e**), and the percentage distribution of organoids with the indicated number of lobes per mIO (**f**) in mIOs from *Lgr5-CreERT2;Cdh1^{fl/fl}* mice treated with or without Amuc_1409*. Scale bar, 200 µm. qRT-PCR results showing the relative mRNA expression of ISC markers (*Lgr5, Olfm4, Msi1, Bmi1,* and *Hopx*), cell

proliferation markers (*Ki67, Pcna,* and *Ccnd1*) (**g**), and Wnt/β-catenin target genes (**h**) in mIOs from *Lgr5-CreERT2;Cdh1^{fl/fl}* mice treated with or without Amuc_1409*. **i** Schematic summary of the role of Amuc_1409 in ISC-mediated intestinal regeneration during irradiated- or 5-FU-induced intestinal injury and aging. The illustration was created with BioRender.com and has been granted a publication license. The chemical structure was created with ChemDraw. All data are presented as the mean ± SEM. In (**a**) and (**e**–**h**), a different symbol indicates a data point representing each biological replicate from independently established organoid lines derived from distinct mouse litters (*n* = 3 biologically independent mice). In (**a, g,** and **h**), each biological replicate includes two or three technical replicates. Statistical analyses were performed using two-tailed Student's *t*-test (**a**) and one-way ANOVA with Dunnett's multiple comparisons test (**e**–**h**) (ns, not statistically significant, *$p < 0.05$, **$p < 0.01$, and ***$p < 0.001$ vs control group). Source data, including the exact *p*-values, are provided as a Source Data file.

SnakeSkin (68100, Thermo Fisher Scientific, Bremen, Germany) at 4 °C for 24 h in Milli-Q water, and then freeze-dried.

### Sample preparation for proteome analysis

For proteomic analysis of secreted proteins from *A. muciniphila*, cells were cultured for 48 h in a total volume of 600 mL. The supernatant was collected from cells cultured in bottles. At first, cells were spun down from the supernatant at $5000 \times g$ for 30 min at 4 °C. To further eliminate any remaining cellular debris, the collected supernatant was

centrifuged at $10,000 \times g$, followed by $20,000 \times g$ for 30 min at 4 °C. Subsequently, the supernatant collected after the final centrifugation at $20,000 \times g$ was subjected to filtration using a 0.22 µm-pore filter (Merck Millipore) to remove the residual bacterial cells and debris. The filtered supernatant was stored at −80 °C until used. The cell-free supernatant of each sample was incubated with Cleanascite™ (Biotech Support Group, North Brunswick, NJ, USA) to remove mucin in the medium. In brief, Cleanascite™ (0.5 mL) was added to the supernatant (1 mL, 1:2 v/v) and mixed for 20 min at room temperature with gentle

rotation. Following centrifugation (16,000 × g) for 2 min at 4 °C, acetone precipitation of the proteins in the supernatant was carried out. Each sample was mixed with cold acetone (−20 °C) to be 1/4 (v/v) and then incubated at −20 °C for 2 h. Samples were spun down for 10 min at 15,000 × g and 4 °C and the protein pellet was washed twice with ice-cold acetone (1 mL). The resulting protein pellets were resuspended with 8 M urea in 50 mM ammonium bicarbonate and the protein concentration was determined using a bicinchoninic acid (BCA) assay.

The proteins from each sample (200 μg) were denatured with 10 mM dithiothreitol (DTT) for 2 h at 37 °C. The reduced cysteine residues were alkylated with 20 mM iodoacetamide (IAA) for 30 min at room temperature in the dark. Excess IAA was quenched with 20 mM L-cysteine for 30 min at room temperature. To reduce the final concentration of urea to 1 M, protein mixtures were diluted with 50 mM ammonium bicarbonate buffer and digested with trypsin (1:50, w/w) for 18 h at 37 °C. The resulting digests were desalted using HLB Oasis cartridges (Waters, Milford, MA, USA), and the eluted peptides were dried in a centrifugal vacuum concentrator (HyperVAC-LITE, Gyrozen, Rep. of Korea). Three biological replicates for each condition were prepared.

### Nanoflow LC-MS/MS (nLC-MS/MS) analysis

Lyophilized peptide samples were dissolved in 0.1% (v/v) formic acid in water and analyzed using a 1260 capillary LC system (Agilent Technologies, Waldbronn, Germany) connected to a Q-Exactive™ Hybrid-Quadrupole Orbitrap mass spectrometer (Thermo Fisher Scientific). Peptides were separated on a reverse phase (RP) column (length, 15 cm; inner diameter, 75 μm) packed with C18 resin (3 μm−100 Å) at a column flow rate of 200 nL/min. RP-nLC gradient elution was carried out with mobile phase A (0.1% (v/v) formic acid in water) and mobile phase B (2% water/0.1% (v/v) formic acid in acetonitrile) for 120 min via the following conditions: 2% B for 10 min, 2−8% B for 1 min, 8−15% B for 1 min, 15−30% B for 70 min, 30−90% B for 3 min, 90% B for 15 min, 90−2% B for 2 min, and 2% B for 15 min.

MS/MS analyses of peptides eluted via RP-nLC run were conducted using a Q-Exactive™ Hybrid-Quadrupole Orbitrap mass spectrometer in data-dependent mode. Precursor ions were surveyed at a resolution of 70,000 within mass ranges of m/z 300−1800. The automatic gain control target value was 3e6. The 12 most intense ions with charge states ≥2 were selected for high-energy collision dissociation fragmentation with normalized collision energy of 27%. MS/MS spectra were acquired at a resolution of 17,500.

### Data analysis

Raw files obtained with LC-MS/MS were analyzed with MaxQuant software version 1.6.6.0 (Max Planck Institute, Munich, Germany) using the Uniprot *A. muciniphila* database (9 March 2021 release; 2137 entries)[56]. The search criteria were as follows: two missed cleavages per peptide were allowed; mass tolerance was set to 4.5 ppm and 20 ppm for precursors and fragment ions, respectively; fixed modifications were carbamidomethylation (C); variable modifications were oxidation (M) and acetylation (N-term); the false discovery rate (FDR) was set to 0.01 at both the peptide and protein levels. Potential contaminants, only identified by site and reverse hits, were filtered out from the list. Signal peptides were predicted using SignalP 5.0 (set for gram-negative bacteria)[57] and SecretomeP 2.0 (set for gram-negative bacteria)[58]. Subcellular localization of proteins was determined using PSORTb 3.0 (set for gram-negative bacteria)[59] and visualized using https://www.bioinformatics.com.cn/en, a free online platform. The abundance of proteins was determined with MS/MS counts and iBAQ (intensity-based absolute quantification) values[60].

### Proteomic analysis of subcellular fractions from *A. muciniphila*

*A. muciniphila* was cultured for 48 h in a basal medium supplemented with 0.2% (w/v) mucin, and PP and OM proteins of *A. muciniphila* were

extracted as previously described[61]. Bacterial cells were spun down from the supernatant at 5000 × g for 30 min at 4 °C. The supernatant was discarded, and the last drops of liquid were carefully removed with a pipette. The cell pellet was gently resuspended with Tris-sucrose-EDTA (TSE) buffer (200 mM Tris-HCl pH 8.0, 500 mM sucrose, 1 mM EDTA) containing a protease inhibitor cocktail (Roche, West Sussex, UK). The suspension was incubated for 30 min at 4 °C and then centrifuged at 16,000 × g for 30 min at 4 °C. For the separation of the OM proteins from the PP proteins, the supernatant was further centrifuged at 100,000 × g for 1 h at 4 °C. From this step, the supernatant containing the PP proteins was collected and the pellet containing the OM fraction was resuspended in PBS containing 0.5% NP-40. The samples from each fraction were stored at −80 °C until used.

Equal amounts (50 μg) of proteins from whole cell lysate, PP fraction, OM fraction and cell-free culture supernatant of *A. muciniphila* were digested with trypsin. The resulting digests from each sample were introduced onto nanoflow liquid chromatography-tandem mass spectrometry (nLC-MS/MS) and followed by MS/MS analyses of tryptic peptides via the same procedures as described in "Nanoflow LC-MS/MS (nLC-MS/MS), Methods". To examine differential expression levels of Amuc_1409 in four subcellular locations (whole cell lysate, PP, OM, cell-free culture supernatant), label-free quantification (LFQ) was conducted using two MS1-based LFQ methods as the MaxQuant software 1.6.6.0 and Proteome Discoverer 3.0 with Minora tool (Sequest HT with Percolator). The relative abundance was calculated as the intensity ratio of each protein to total protein in each fraction sample.

### Expression and purification of His-tagged Amuc_1409 and His-tagged Amuc_1100 protein

The Amuc1409 and Amuc1100 genes were amplified from the genomic DNA of *A. muciniphila* with PCR using the primers. The Amuc1409 gene was amplified using primers 5′-ATA TAC ATA TGA TCC CGG AGT CTT CCG TTC ATA TG-3′ and 5′-TGG TGC TCG AGA CGG TCC ACA TGA AGC TCG AG-3′, while the Amuc1100 gene was amplified using primers 5′- GGG TAC CAT ATG ATC GTC AAT TCC AAA CGC-3′ and 5′-CCT GGC TCG AGA TCT TCA GAC GGG TTC CTG-3′. For the production of His-tagged recombinant Amuc_1409 and Amuc_1100 proteins, expression plasmids with kanamycin resistance genes were constructed. The target genes, devoid of the coding sequences for their signal sequences, were amplified and cloned into the pET-30a vector (Lucigen, Middleton, WI, USA) using *E. coli* DH5α. The resulting plasmids, pET30a-1409 and pET30a-1100, were verified by DNA sequence analysis, and the purified plasmids containing the recombinant Amuc_1409 and Amuc_1100 genes were transformed into *E. coli* BL21 (DE3) (Lucigen). The transformed *E. coli* strains were grown in LB broth supplemented with kanamycin (50 μg/mL) at 30 °C with shaking at 120 rpm. During the mid-exponential phase, 0.5 mM IPTG was added to induce protein expression. The cells were harvested after centrifugation at 6600 × g for 20 min at 4 °C. Next, the cell pellets were resuspended in an ice-cold buffer containing 300 mM NaCl, 50 mM Tris-HCl (pH 8.0), and 10 mM imidazole. The cell suspensions were sonicated, and the crude cell extracts were obtained. The crude cell extracts were further processed by centrifugation at 11,000 × g for 30 min at 4 °C. The resulting cell lysates containing the 6 His-Amuc1409 and His-Amuc1100 fused proteins were applied to Ni Sepharose 6 Fast Flow columns (Cytiva, Uppsala, Sweden). Endotoxin was removed from the recombinant proteins using High Capacity Endotoxin Removal Spin Columns (Thermo Fisher Scientific, Waltham, MA, USA) following the manufacturer's instructions. After buffer exchange using 7 MWCO Zeba Spin Desalting Columns (Thermo Fisher Scientific), the protein contents were determined using the BCA assay (Thermo Fisher Scientific). The purified Amuc_1409 and Amuc_1100 proteins were then stored at −70 °C.

## Expression and purification of His-tagged P9 protein

The P9 gene was also amplified from the genomic DNA of *A. mucini-phila* by PCR using primers carrying the NdeI and XhoI sites: 5′- AAG GAG ATA TAC ATA TGA ACA TGC ACT CAT TCC GTT G-3′ and 5′-GGT GGT GGT GCT CGA GTT TTC CGG AGG ATT CCA GC-3′. Similarly, for the production of His-tagged recombinant P9 protein, an expression plasmid containing a kanamycin resistance gene was constructed by amplifying the gene of interest, devoid of the coding sequence for its signal sequence, and cloning the resulting PCR product into the pET-30a vector using *E. coli* DH5α strain (Lucigen, USA). The resulting plasmids were transformed into the *E. coli* BL21 (DE3) strain, and the cells were grown overnight at 37 °C in 20 mL Luria-Bertani (LB) broth containing 100 μg/mL kanamycin. The culture cells were resuspended in 2 L of the same media and grown at 37 °C to an $OD_{600}$ of 0.5-0.6. His-tagged recombinant protein was then induced using 0.5 mM isopropyl-β-Dthiogalactopyranoside (IPTG; LPS Solution, Daejeon, Korea). After further incubation at 18 °C for 20 h, cells were harvested by centrifugation at $4000 \times g$ for 5 min, and recombinant proteins were purified using a typical purification procedure. Briefly, harvested cells were resuspended in lysis buffer (50 mM Tris-HCl, pH 7.5, 300 mM NaCl, and 30 mM imidazole), lysed by high-pressure homogenizer, and centrifuged at $14,000 \times g$ at 4 °C for 1 h. The supernatant was subjected to immobilized metal affinity chromatography (IMAC) using nickel-nitrilotriacetic acid (Ni-NTA) resin (Qiagen, Hilden, Germany) pre-equilibrated with lysis buffer. The column was washed with 20 column volumes (1 L) of wash buffer containing (50 mM Tris-HCl, pH 7.5, 300 mM NaCl, and 50 mM imidazole) and His-tagged recombinant proteins bound to resin were eluted with elution buffer (50 mM Tris-HCl, pH 7.5, 300 mM NaCl, and 250 mM imidazole). The endotoxin and buffer exchange were performed using the same method and then stored at −70 °C.

## Expression and purification of His-tag cleaved 1409 protein

To express the His-tag cleaved 1409 protein, the Amuc1409 gene was amplified from the genomic DNA of *A. muciniphila* by PCR using the modified primers. The forward primer was designed with the NdeI site replaced by an EcorI site (5′-ATA TAG AAT TCA TCC CGG AGT CTT CCG TT-3′), while the reverse primer, carrying the XhoI site, remained unchanged. The amplified products and a pProEX HTa vector (Invitrogen) containing N-terminal hexahistidines-tobacco etch virus (His-TEV) were digested using EcoRI and XhoI. The amplified 1409 products were ligated into the pProEX HTa vector and verified by DNA sequencing. The resulting plasmids was transformed into the *E. coli* BL21 (DE3) strain, and the cells were grown overnight at 37 °C in 20 mL Luria-Bertani (LB) broth containing 100 μg/mL ampicillin. The culture cells were resuspended in 2 L of the same media and grown at 37 °C to an $OD_{600}$ of 0.5−0.6. His-tagged recombinant protein was then induced using 0.5 mM isopropyl-β-Dthiogalactopyranoside (IPTG; LPS Solution, Daejeon, Korea). After further incubation at 18 °C for 20 h, cells were harvested by centrifugation at $4000 \times g$ for 5 min, and recombinant proteins were purified using a typical purification procedure. Briefly, harvested cells were resuspended in lysis buffer (50 mM Tris-HCl, pH 7.5, 300 mM NaCl, and 10 mM imidazole), lysed by high-pressure homogenizer, and centrifuged at $14,000 \times g$ at 4 °C for 1 h. The supernatant was subjected to immobilized metal affinity chromatography (IMAC) using nickel-nitrilotriacetic acid (Ni-NTA) resin (Qiagen, Hilden, Germany) pre-equilibrated with lysis buffer. The column was washed with 20 column volumes (1 L) of wash buffer containing (50 mM Tris-HCl, pH 7.5, 150 mM NaCl, and 30 mM imidazole) and His-tagged recombinant proteins bound to resin were eluted with elution buffer (50 mM Tris-HCl, pH 7.5, 150 mM NaCl, and 150 mM imidazole). N-terminal His-tags were removed from recombinant proteins when necessary by treating with TEV protease for 20 h at 4 °C during dialysis in buffer containing 50 mM Tris-HCl, pH 7.5 and 150 mM NaCl. Non-digested proteins by TEV proteases were removed

from samples by a second affinity chromatography step using Ni-NTA resin, and eluted proteins were further purified by gel-filtration chromatography using a HiLoad 16/60 Superdex™ 75 prep grade column (GE Healthcare Life Sciences, Chicago, IL). The purified 1409 in 1 x PBS was concentrated at 7.5 mg/mL.

## Experimental mice

C57BL/6J, B6.129P2-*Lgr5*[tm1(cre/ERT2)Cle]/J, and B6.129-*Cdh1*[tm2/Kem]/J mice were purchased from The Jackson Laboratory (Bar Harbor, ME, USA). BALB/c mice were purchased from Orient Bio Inc. (Seongnam, Rep. of Korea). All mice were maintained at the Korea Research Institute of Bioscience and Biotechnology (Daejeon, Rep. of Korea). *Lgr5-CreERT2;Cdh1*[fl/fl] mice were generated by cross-breeding B6.129P2-*Lgr5*[tm1(cre/ERT2)Cle]/J and B6.129-*Cdh1*[tm2/Kem]/J mice. All mice were housed under a constant 12-h light/dark cycle at a suitable temperature ($22 \pm 2$ °C) and humidity (typically $55 \pm 5$%) with a standard chow diet (Teklad Global 18% Protein Rodent Diet 2018S; 18.6% crude protein, 6.2% fat, 44.2% available carbohydrates; Envigo, Indianapolis, IN, USA) and water provided *ad libitum* in a specific pathogen-free facility. Mice between 2 and 4 months of age were considered young and mice over 22 months of age were considered old. 100-week-old male C57BL/6 mice were treated orally with vehicle or Amuc_1409* (4.5 μg per mouse) daily for 15 weeks. All animal experiments were approved by the Institutional Animal Care and Use Committee of the Korea Research Institute of Bioscience and Biotechnology (KRIBB-AEC-22244, KRIBB-AEC-22193) and were performed in accordance with the Guide for the Care and Use of Laboratory Animals published by the US National Institutes of Health.

## Irradiation and BrdU administration

For irradiation, 10-week-old male C57BL/6J mice were exposed to single total body irradiation of 1100 Rad using a Gamma cell 3000 (Nordion Inc., Ottawa, Canada) at the KAIST Laboratory Animal Resource Center (Daejeon, Rep. of Korea). After irradiation, mice were treated orally with vehicle or Amuc_1409* (9 μg per mouse) daily for 8 days for survival analysis (Figs. 4a) and for 5 days for intestinal tissue isolation (Fig. 4c). For BrdU staining, mice were injected intraperitoneally 6 h before euthanasia with 200 μL of BrdU (Sigma-Aldrich) dissolved in PBS to 5 mg/mL, and intestinal tissue samples were collected at day 5 post-irradiation.

## Treatment of 5-FU and assessment of body weight and diarrhea

10-week-old male C57BL/6J mice were given a single intraperitoneal injection of 5-FU (Sigma-Aldrich, 50 mg/kg/day) daily for 4 days and were treated orally with vehicle or Amuc_1409* (9 μg per mouse) daily for 6 days following the first 5-FU injection. Body weight and a diarrhea score for each mouse were recorded daily. For diarrhea assessment, mice were placed in individual cages having a paper towel placed on their floor and each mouse stool was collected and scored. The severity of the diarrhea was scored using the following scale[62,63]: 0: normal (normal stool), 1: minimal (soft stool), 2: slight (slightly wet and soft stool), 3: moderate (wet and slightly unshaped stool with watermark beyond the edge of the stool on the paper towel), 4: severe (watery and unshaped stool with watermark beyond the edge of the stool on the paper towel). Mice were sacrificed via cervical dislocation on day 6 and intestinal tissue samples were collected and stored in 10% neutral buffered formalin (NBF) and at −80 °C for histological and quantitative real-time polymerase chain reaction (qRT-PCR) analyses, respectively.

## Tumor inoculation and detection

To establish CT-26 syngeneic mouse model, $0.5 \times 10^6$ CT-26 cells in 100 μL PBS were subcutaneously injected into the right flank of 8-week-old female BALB/c mice. At 7 days after tumor cell inoculation, tumor-bearing mice were orally administered either vehicle or

Amuc_1409* (9 µg per mouse) for a duration of 14 days. Alteration in the body weight and tumor volume of each mouse was observed every day for 14 days after the treatment. Briefly, the length (L) and width (W) of tumors were measured using a digital vernier caliper (Matusutoyo, Tokyo, Japan), and the volume of each tumor was calculated using the following formula: Tumor volume (mm$^3$) = (W$^2$ × L)/2. At 14 days after vehicle or Amuc_1409* treatment, all mice were sacrificed and the tumors were excised and weighed.

## Protein extraction from mouse whole fecal samples

To measure the physiological level of the Amuc_1409 protein in the mouse gut, protein extraction from mouse whole fecal samples was carried out as previously described[64]. We collected stools, intestinal contents, and intestinal wall samples from the small intestine, cecum, and large intestine of each 12-week-old male C57BL/6 mouse. Intestinal contents from the small intestine, cecum, and large intestine were flushed out with up to 30 mL of ice-cold PBS with a protease inhibitor cocktail (Roche). The whole fecal samples in the PBS were dispersed by applying pressure with a 3 mL syringe plunger and then transferred to a 50 mL tube. After centrifugation at 2000 × g for 10 min at 4 °C, the supernatant was discarded, and the pellets were resuspended in freshly prepared PBS with a protease inhibitor cocktail at a concentration of 1 g/mL. The pellets were immersed in liquid nitrogen for 1 min and then thawed at room temperature for 20 min, followed by vortexing at maximum speed for 5 s. This freeze-thaw-vortex process was repeated three times. Subsequently, the contents were sonicated for 5 cycles at 20% amplitude with 30 s on/off pulse on ice and were then transferred to new 1.5 mL microtubes. The samples were centrifuged at 14,000 × g for 30 min at 4 °C. The protein concentration in the supernatant was measured using the Bradford assay.

## Histological analysis

At necropsy, small intestine samples were immediately fixed in 10% NBF, embedded in paraffin, and cut into 4-µm-thick slices. The slides were stained with H&E. Then, H&E sections were analyzed microscopically, and histopathological scores were determined in a blinded manner, using the following criteria as previously described[65]: epithelial surface erosion, distortion of the villous, distortion of crypt architecture, and reduced villous height. The score was graded according to severity as 0 for normal, 1 for mild, and 2 for severe for each parameter. Data are shown as the mean of the sum of scores obtained for four parameters for each small intestine.

## qRT-PCR

Total RNA was isolated from the small intestine tissue or mouse small intestinal organoids using Trizol reagent (Invitrogen, Carlsbad, CA, USA), and reverse-transcribed using the iScript™ cDNA Synthesis kit (BioRad, Hercules, CA, USA). The resulting cDNA was subjected to qPCR using the StepOnePlus™ Real-Time PCR System (Applied Biosystems, Foster City, CA, USA) with AccuPower® 2X Greenstar qPCR Master Mix (Bioneer, Daejeon, Rep. of Korea), according to the manufacturer's protocol. Relative gene expression levels were analyzed using the 2$^{(-\Delta\Delta Ct)}$ method and normalized relative to 18 S rRNA expression.

The gene expression analysis in the human intestinal organoids was conducted as follows: total RNA was prepared from human intestinal organoids (hIOs) using an RNeasy Mini Kit (Qiagen, Valencia, CA, USA), and cDNA was synthesized using the Superscript IV cDNA Synthesis Kit (Thermo Fisher Scientific). qPCR analysis was performed using a 7500 Fast Real-time PCR system (Applied Biosystems, Foster City, CA, USA) according to the manufacturer's instructions. Relative gene expression levels were analyzed using the 2$^{(-\Delta\Delta Ct)}$ method and normalized relative to GAPDH expression. The primer sequences used in the experiments are listed in Supplementary Table 7.

## Crypt isolation and culture of organoids derived from mouse small intestines

Mouse small intestines were cut open longitudinally and swirled in cold PBS in a Petri dish for rinsing. The tissue was cut into 1–2 mm pieces, which were pipetted up and down approximately 10 times using a 10-mL serological pipette pre-wetted with PBS. After the pieces were settled by gravity for approximately 30 s, the supernatant was removed, and cold PBS (10 mL) was added. This process was repeated 20–25 times until the supernatant was clear. Next, the tissue fragments were incubated in gentle cell dissociation reagent (25 mL; STEMCELL Technologies, Vancouver, Canada, ST07174) for 20 min at room temperature on a rocking platform. After the fractions were settled by gravity, the supernatant was removed, and cold PBS containing 0.1% (w/v) bovine serum albumin (BSA) (10 mL) was added and pipetted up and down using a 10-mL serological pipette. The supernatant was passed through a 70-µm cell strainer, and the crypt fraction was collected. After the crypt purity of each fraction was evaluated under a microscope, the fraction enriched for intestinal crypts was selected and centrifuged at 270 × g for 5 min at 4 °C. The supernatant was discarded, and the pellet was resuspended in Advanced DMEM/F-12 (10 mL; Thermo Fisher Scientific) for washing and centrifuged at 270 × g for 5 min at 4 °C. The pelleted crypts were mixed with Matrigel (Corning Inc., Corning, NY, USA) and crypt culture medium (IntestiCult™ Organoid Growth Medium, STEMCELL Technologies) at a 1:1 ratio, and the suspended crypts (50 µL) were plated in a pre-warmed 24-well plate. After allowing the Matrigel to polymerize for 20 min, 500 µL of crypt culture medium was added per well, and the plate was placed in a humidified incubator (5% CO$_2$) at 37 °C. The organoids were mechanically passaged 7 days after plating, and at least 50 crypt fragments were seeded per 24-well plate for experiments. The number of all organoids per well and the de novo crypt (lobes) formation number per organoid were manually counted under light microscopy, with at least individual 2 wells analyzed for each treatment group.

For Cdh1 deletion in Lgr5$^+$ stem cells, the organoids generated from Lgr5-CreERT2;Cdh1$^{fl/fl}$ mice were treated with 4-hydroxytamoxifen (4-OHT, 1 µM, Selleck Chemicals, Houston, TX, USA) for 6 consecutive days.

## Differentiation and culture of hIOs derived from human pluripotent stem cells (hPSCs)

This research received approval from the Korean Public Institutional Review Board (IRB numbers: P01-201409-ES-01-09, P01-201609-31-002). hIOs were generated through the differentiation of the H9 (WA09, XX; WiCell Research Institute, Madison, WI) or H1 (WA01, XY; WiCell Research Institute) hESC line and a hiPSC line derived from CRL-2097 (CCD-1079SK, skin fibroblast; ATCC, Rockville, MD, USA), following established protocols[66,67]. hPSC-derived hIOs were generated as described previously[66]. Briefly, the differentiation of hPSCs into definitive endoderms (DEs) was induced by treatment with 100 ng/mL activin A (R&D Systems, Minneapolis, MN, USA) for 3 days in RPMI1640 (Thermo Fisher Scientific) containing 0, 0.2, and 2% (v/v) fetal bovine serum (FBS, Thermo Fisher Scientific). The hindgut (HG) spheroids were organized by addition of 500 ng/mL FGF4 (Peprotech, Rocky Hill, NJ, USA) and 3 µM CHIR99021 (Tocris, Ellisville, MO, USA) in DMEM/F12 (Thermo Fisher Scientific) containing 2% (v/v) FBS for 4 days. The HG spheroids were gently embedded in Matrigel dome and cultured in advanced DMEM/F12 (Thermo Fisher Scientific) containing 1× B27 supplement, 100 ng/mL EGF (R&D Systems), 500 ng/mL R-spondin1 (Peprotech), and 100 ng/mL Noggin (R&D Systems). To evaluate the effect of Amuc_1409 on the growth of hIOs, hIOs were treated with various concentrations (4, 8, 16 nM) of Amuc_1409* every other day for two passages. We counted the number of lobes per hIO and calculated the surface area using the horizontal cross-section of hIOs to determine the size of hIOs.

## IHC and IF staining

Paraffin-embedded sections of the small intestines from mice were deparaffinized in xylene and rehydrated with graded alcohols. After antigen retrieval through boiling in citrate buffer (pH=6), endogenous peroxidase activity was quenched using 3% hydrogen peroxide for 30 min, followed by blocking sections in normal goat serum. The sections were incubated with a monoclonal rabbit anti-Ki67 antibody (Abcam, Cambridge, UK) or anti-OLFM4 antibody (Cell Signaling Technology, Danvers, MA, USA) at 4 °C overnight. The sections were incubated with biotinylated goat anti-rabbit IgG secondary antibody (Vector Laboratories, Burlingame, CA, USA) at room temperature for 1 h, followed by incubation with avidin-biotinylated horseradish peroxidase (HRP) complex (Vectastain Elite ABC-HRP kit, PK-6101, Vector Laboratories) at room temperature for 30 min. The signals were developed with the 3,3′-diaminobenzidine (DAB) chromogen (DAB Peroxidase HRP Substrate Kit; SK-4100; Vector Laboratories), and cell nuclei were labeled with haematoxylin. Then, histological sections were viewed at 400× magnification and images were obtained using a microscope (Olympus BX51, Tokyo, Japan). The average number of Ki67- or OLFM4-positive cells per crypt in the small intestine was determined by counting approximately 30 well-preserved crypts per mouse.

For IF staining of mIOs, the mIOs cultured in collagen-coated chamber slides (LabTek, Nunc, Roskilde, Denmark) were fixed with 4% (v/v) paraformaldehyde (Sigma-Aldrich) for 20 min and incubated for 20 min with 100% methanol at −20 °C. The fixed organoids were blocked with 2% (v/v) FBS diluted in PBS for 1 h, incubated with primary antibodies (Supplementary Table 8) for 12 h at 4 °C, and then with corresponding Alexa Fluor-conjugated secondary antibodies (Thermo Fisher Scientific) for 1 h at room temperature. Diamidino-phenylindole (DAPI; 1 μg/mL) was used to visualize the nuclei, and then the slides were mounted with mounting solution. Images were acquired under a Nikon laser scanning confocal microscope (C2plus, Nikon, Tokyo, Japan).

IF staining of hPSC-derived hIOs was performed as described previously[67]. hPSC-derived hIOs were fixed in 4% (v/v) paraformaldehyde. After fixation, hIOs were cryo-protected in 10–30% (w/v) sucrose solution. Then, the hIOs were embedded in optimal cutting temperature compound (Sakura Finetek, Tokyo, Japan) and frozen. The samples were sectioned at a thickness of 15 μm using a cryostat microtome. hIO sections were permeabilized with 0.1% (v/v) Triton X-100 and blocked with 4% (w/v) BSA. The samples were incubated with primary antibodies (Supplementary Table 8) at 4 °C overnight and with secondary antibodies at room temperature for 1 h. DAPI was used for nuclear staining. Samples were examined with a fluorescence microscope (IX51, Olympus, Japan) and EVOS FL Auto 2 (Thermo Fisher Scientific).

## Construction of plasmids

Full-length, extracellular, and intracellular domains of E-cadherin were amplified with PCR from a pCMV3-HA-E-cadherin plasmid (HG10204-NY, Sino Biological, Beijing, China) using primers listed in Supplementary Table 9. The final PCR product was digested with SacII and XhoI and ligated into the Strep-tag vector pEXPR-IBA105 (IBA BioTAGnology, IBA US Distribution Center, St. Louis, MO, USA).

## Cell culture and transient transfection

Human embryonic kidney 293 T (HEK293T; ATCC® CRL-11268) and HT-29 (ATCC® HTB-38) cells were cultured in Dulbecco's modified Eagle's medium (DMEM; Welgene Inc., Rep. of Korea) containing 10% (v/v) FBS (Gibco, Thermo Fisher Scientific), 100 units/mL penicillin, and 100 μg/mL streptomycin (Gibco) in a humidified environment (5% CO₂/95% air) at 37 °C. HEK293T cells were seeded at $3 \times 10^6$ cells per well in 100-mm dishes. After 18 h of culture, cells were transiently transfected with the indicated plasmids using Lipofectamine LTX & Plus Reagent (Invitrogen) according to the manufacturer's instructions. The cells were harvested 24–36 h post-transfection.

## Immunoprecipitation and in vitro binding assays

The HT-29 or HEK239T cells were washed with PBS and lysed in RIPA lysis buffer (1% (v/v) NP-40, 0.25% (v/v) sodium deoxycholate, 50 mmol/L Tris-HCl pH 7.4, 1 mmol/L EDTA, and 120 mmol/L NaCl, including protease and phosphatase inhibitors). Insoluble material was removed by centrifugation at $14,000 \times g$ at 4 °C for 3 min. To pre-clear the unspecific proteins, protein A/G PLUS-agarose (20 μL; Santa Cruz Biotechnology, Santa Cruz, CA, USA) was added to the whole-cell lysates (WCLs) and incubated for 2 h at 4 °C with gentle agitation. The pre-cleared WCLs were incubated overnight with 2 mg of target protein antibodies or Strep-Tactin beads (IBA Life Sciences, Gottingen, Germany) at 4 °C with gentle agitation. For the WCLs incubated with antibodies, protein A/G PLUS-agarose (20 μL) was added to precipitate the proteins attached to antibodies and incubated for 4 h at 4 °C with agitation. After centrifugation, the immunoprecipitates were collected and washed with washing buffer.

For the in vitro binding assay, immunoprecipitated Strep-tagged proteins were prepared from lysates of HEK293T cells transfected with individual plasmids. Immunoprecipitates were incubated with Amuc_1409* protein (20 μg) in binding buffer (25 nM HEPES, 100 mM NaCl, 0.01% (v/v) Triton X-100, 5% (v/v) glycerol, 1 mM DTT) in a final volume of 800 μL for 3 h at 4 °C with gentle agitation. After washing with wash buffer (150 mM NaCl, 50 mM Tris-HCl, pH 6.8, and 1% NP40), immunoprecipitates were dissolved in SDS sample buffer and separated with 4–12% SDS-PAGE.

## Preparation of subcellular fractions

HT-29 cells were treated with 16 nM Amuc_1409* for 30 min followed by the extraction of membrane and cytosolic protein fractions. Membrane and cytosolic proteins were prepared with the Mem-PER™ Plus Membrane Protein Extraction Kit (Thermo Fisher Scientific, #89842) according to the manufacturer's instructions.

## Western blot analysis

Protein samples were separated using SDS-PAGE and transferred onto a PVDF membrane (Millipore). Membranes were incubated with target primary antibodies (Supplementary Table 8) in Tris-buffered saline containing 0.02% (v/v) Tween 20 (TBST) and 3% (w/v) BSA overnight at 4 °C. After washing the membranes three times with TBST for 10 min, the membranes were incubated with horseradish peroxidase-conjugated secondary antibodies for 1 h at room temperature. After the washing step was repeated, bands were detected using EzWestLumi plus (ATTO, Tokyo, Japan). In Fig. 7d, densitometric analysis of the protein band was performed using the ImageJ v.1.43 u software (National Institutes of Health, USA). The quantification data are expressed as relative densitometer units with respect to the control group of each fraction (RDU/C). The data were calculated by dividing the band signal intensity of each band corresponding to the protein of interest by the value of the band corresponding to the protein used as loading control of each fraction, obtained from the same sample in the same membrane. This result, in turn, was divided by the value obtained from the control group of each fraction. We have provided uncropped and unprocessed scans of the most important blots in the Source Data file.

## Circular dichroism

28.5 μM Amuc_1409 proteins were prepared in PBS buffer (pH 7.3) for thermal unfolding experiments. CD data were collected on a Jasco J-815 CD spectrometer (Tokyo, Japan) with a 1-mm path-length cuvette. The CD spectrum was monitored at 216 nm by raising the temperature in 1 °C intervals from 10 to 100 °C. CD thermal melt data was normalized to a two-state unfolding model[68,69] by the standard equation: Fraction of native 1409 = $(\theta_t - \theta_{100})/(\theta_{10} - \theta_{100})$. In this equation, $\theta_{10}$ and $\theta_{100}$

represent the ellipticity for fully folded and unfolded forms, respectively; $\theta_t$ represents the ellipticity at any temperature. The melting temperature ($T_m$) was calculated by a non-linear curve fit of Boltzmann method using Origin-v8.0 program (Origin Lab Corporation, USA).

## Statistics and reproducibility

Statistical analyses were conducted using GraphPad Prism v8 software (GraphPad, San Diego, CA, USA). Comparisons between groups were performed using a two-tailed Student's $t$-test (for comparison of two experimental conditions) or analysis of variance (ANOVA) (for comparison of three or more experimental conditions). The survival ratio was analyzed with the Log-Rank test. All data are presented as the mean ± standard error of the mean (SEM) and results with $p$ values < 0.05 were considered statistically significant. All experiments were replicated as indicated in the respective figure legends and all replications showed reproducibly similar results. No statistical method was used to predetermine sample size. Sample sizes were determined based on $n$ values needed to evaluate differences between groups in prior published studies or our previous experiences. No data were excluded from the analyses. For in vivo experiments, age- and sex-matched mice were randomly allocated to each experimental group, ensuring an equal distribution of body weight among the groups. For in vitro and ex vivo experiments, cell and organoid cultures were randomly assigned to each experimental group. Although we were unable to blind the experimental design, genotypes of the animals, and group allocation during experiments, in general, data acquisition and analysis were conducted in a blinded manner and confirmed by multiple investigators. Especially, for histopathological and microscopical examinations (histological scoring, IHC/IF analysis, and quantification of organoid size and lobe number), samples were blinded to ensure unbiased imaging and analyses.

## Reporting summary

Further information on research design is available in the Nature Portfolio Reporting Summary linked to this article.

## Data availability

The MS-based proteomic datasets generated in this study have been deposited to the ProteomeXchange Consortium via the PRIDE partner repository under accession code PXD042323. All data generated and/or analyzed during this study are included in this published article and its Supplementary information/Source Data files and available from the corresponding author on reasonable request. Amuc_1409 protein is available upon request under a material transfer agreement (MTA) with the Korea Research Institute of Bioscience and Biotechnology (KRIBB). Source data are provided with this paper.

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

## Acknowledgements

This work was supported by the Bio & Medical Technology Development Program (2019M3A9F3065867; C.-H.L.) of the National

Research Foundation of Korea (NRF) funded by the Ministry of Science and ICT, and by the KRIBB Research Initiative Program (KGS1062423; C.-H.L.).

## Author contributions

Y.-H.K., D.K. and C.-H.L. conceived and supervised the study. E.-J.K., J.-H.K., Y.E.K., Y.-H.K., D.K. and C.-H.L. designed the all experiments. E.-J.K. and J.-H.K. carried out all experiments for animal, mouse intestinal organoids, and molecular biology, and analyzed and interpreted the relevant data under the supervision of Y.-H.K. and C.-H.L. M.-Y.R., J.G., J.H.C., Y.-K.C., I.-B.L., D.-H.C., Y.-J.S. and J.-R.N. provided technical assistance for animal experiments. H.L., K.B.J., S.P. and Y.H. performed all experiments for human intestinal organoids and analyzed and interpreted the relevant data under the supervision of M.-Y.S. Y.E.K., S.S., J.-S.J., H.-J.K. and H.M.Y. performed all experiments for proteomic analysis of secreted proteins from *A. muciniphila* and analyzed and interpreted the relevant data under the supervision of T.-Y.K. and D.K. S.P. and E.-Y.L. provided technical assistance for sample preparation related to proteomic analysis. D.-H.C. performed anaerobic culture experiments using *A. muciniphila*, and D.-H.C., Y.L., E.-J.L. and H.B.K. performed cloning and purified proteins. K.-S.K., J.H.H., Y.-G.K., D.-H.L., H.-J.K., M.H.K., B.-C.K., Y.-H.K., D.K. and C.-H.L. provided assistance with data interpretation and critical scientific discussion. E.-J.K., J.-H.K., Y.E.K., H.L., K.B.J., D.-H.C. and Y.L. drafted the manuscript, and Y.-H.K., D.K. and C.-H.L. edited the manuscript. All authors have read and approved the final manuscript.

## Competing interests

E.-J.K., J.-H.K., D.-H.C., J.G., J.H.C., Y.-K.C., I.-B.L., D.-H.C., Y.-J.S., J.-R.N., K.-S. K., B.-C. K., Y.-H.K. and C.-H.L. are inventors on patent applications dealing with the use of Amuc_1409 and its components in the treatment of different diseases. B.-C.K. is a founder of HealthBiome Inc. The other authors declare no competing interests.

## Additional information

[1]Laboratory Animal Resource Center, Korea Research Institute of Bioscience and Biotechnology (KRIBB), Daejeon 34141, Republic of Korea. [2]Department of Veterinary Pathology, College of Veterinary Medicine, Chungnam National University, Daejeon 34134, Republic of Korea. [3]Livestock Products Analysis Division, Division of Animal health, Daejeon Metropolitan City Institute of Health and Environment, Daejeon 34146, Republic of Korea. [4]Group for Biometrology, Korea Research Institute of Standards and Science (KRISS), Daejeon 34113, Republic of Korea. [5]School of Earth Sciences & Environmental Engineering, Gwangju Institute of Science and Technology (GIST), Gwangju 61005, Republic of Korea. [6]Stem Cell Convergence Research Center, Korea Research Institute of Bioscience and Biotechnology (KRIBB), Daejeon 34141, Republic of Korea. [7]Microbiome Convergence Research Center, Korea Research Institute of Bioscience and Biotechnology (KRIBB), Daejeon 34141, Republic of Korea. [8]Biotherapeutics Translational Research Center, Korea Research Institute of Bioscience and Biotechnology (KRIBB), Daejeon 34141, Republic of Korea. [9]Synthetic Biology Research Center, Korea Research Institute of Bioscience and Biotechnology (KRIBB), Daejeon 34141, Republic of Korea. [10]Laboratory Animal Resource Center, Korea Advanced Institute of Science and Technology (KAIST), Daejeon 34141, Republic of Korea. [11]Department of Bio-Molecular Science, Korea Research Institute of Bioscience and Biotechnology (KRIBB) School of Bioscience, Korea University of Science and Technology (UST), Daejeon 34141, Republic of Korea. [12]Department of Functional Genomics, Korea Research Institute of Bioscience and Biotechnology (KRIBB) School of Bioscience, Korea University of Science and Technology (UST), Daejeon 34141, Republic of Korea. [13]Department of Measurement Science, Korea Research Institute of Standards and Science (KRISS) School of Precision Measurement, Korea University of Science and Technology (UST), Daejeon 34113, Republic of Korea. [14]Department of Applied Biological Engineering, Korea Research Institute of Bioscience and Biotechnology (KRIBB) School of Biotechnology, University of Science and Technology (UST), Daejeon 34141, Republic of Korea. [15]Department of Biosystems and Bioengineering, Korea Research Institute of Bioscience and Biotechnology (KRIBB) School of Biotechnology, University of Science and Technology (UST), Daejeon 34141, Republic of Korea. [16]HealthBiome Inc., Daejeon 34141, Republic of Korea. [17]These authors contributed equally: Eun-Jung Kang, Jae-Hoon Kim, Young Eun Kim. ✉e-mail: yhoonkim@kribb.re.kr; djkang@kriss.re.kr; chullee@kribb.re.kr

