## [Peer Review File · Nature Communications]

The secreted protein Amuc_1409 from *Akkermansia muciniphila* improves gut health through intestinal stem cell regulationEditorial Note: This manuscript has been previously reviewed at another journal. This document only contains reviewer comments and rebuttal letters for versions considered at *Nature Communications*.

Reviewers Comments:

Reviewer #1 (Remarks to the Author):

This revised version of the manuscript on Amuc_1409 has been greatly improved. The authors also provided considerably new information that is now mainly in the detailed responses to the reviewers. They also removed some erroneous conclusions on the mucin induction.

However, a few points remain and new ones popped up:

1. The amount of Amuc_1409 in intestinal A.muciniphila cells that is signaling-competent should be addressed more quantitatively: the protein when purified signals but how much is need for signal transduction ... do the authors see signaling with whole cells? These important points should be also included in the manuscript like Fig R7. Here the cells are grown in basal medium – what then about mucin as this was used to grow the cells initially. I also suggest to include Fig R7 in the revised manuscript.
2. The authors suggest that the Amuc_1409 protein is localized in the periplasm and outer membrane and conclude this from Fig R2 (see below). This is not correct as in this Fig the amounts of loaded material are not provided – so the Amuc_1409 protein may be present but in homeopathic amounts (like in the OM) – this should be addressed and conclusions should be supported by quantitative data – mass spectroscopy based proteome analysis is a better way to quantify.
3. The authors provide data on thermal stability and these are very useful – I suggest to include this in the manuscript as highly relevant.
4. The authors now show the Amuc_1409 protein to be a dimer – is dimerization needed for activity ?
5. Do the authors have any data on the Amuc_1409 stability ? I liked the experiment of Fig R18 and suggest to include this in the revised manuscript. However, the revised title is misleading as all experiments are in mice...so human can be deleted from the new title.

Reviewer #2 (Remarks to the Author):

The revised manuscript goes a good way in addressing the issues raised during the initial submission, and several aspects of the data are now much better controlled and convincing. Regarding the revised manuscript:

1. Staining for secretory cell markers and qPCR (ED Fig. 2) is convincing in showing that Amuc_1409 drives secretory cell differentiation, but there remains no targeted analysis of enterocyte differentiation. Muc3 (also called Muc17) is a marker of mature enterocytes and is induced by treatment, indicating that Amuc_1409 drives general rather than targeted differentiation but this requires validation by examination of other mature enterocyte marker genes (see PMID: 29144463 for options).
2. Data concerning the interaction of Amuc_1409 and E-cadherin (Fig. 4) is now much more convincing; however the current data only supports a role for in Amuc_1409 in driving E-cadherin dependent changes in stem cells and proliferation. Differentiation markers should also be examined (qPCR, IF) to demonstrate that the claimed effects on differentiation are also lost in E-cadherin KO organoids.
3. Control data using different Amuc_1409 constructs (Fig. R13) should be incorporated into the manuscript.

Reviewer #3 (Remarks to the Author):

The authors have addressed a number of my comments. However, there are still some major remarks that in my opinion need clarification. Without this clarification, it is not possible to judge the validity of a number of conclusions obtained.

Major remarks

1) The authors answered to my Q1 that they have included information regarding experimental N numbers and reproducibility. However, it is still not clear to me looking at the majority of the figures/panels how the data points presented represent technical or biological replicates, and whether these are from independent experiments. Additionally, the authors should define their consideration of technical and biological replicates, in order to have a proper view on the sufficiency of the data included here. Are technical replicates from independent experiments and biological replicates from organoid lines derived from different mice/humans? Or technical replicates are the same organoid sample measured 3 times and biological replicate independent experiment with the same organoid line? Are technical replicates measured at different organoid passages? Or the experiments are performed with several wells from same passage, simultaneously, pooling some wells together and using that as replicate? Can the authors clarify the meaning of "RNA pooled from n=12 individual wells per group" and how it relates to the replicates? Is a technical replicate the same RNA sample from these 12 pooled wells ran in 3 different wells of the qPCR plate? I think it is important to clarify the N number in all the plots shown in the manuscript due to the small effect size on mRNA level observed along the manuscript and the small variability (depicted as SEM) observed between replicates. In organoids, especially from different biological replicates/donors the variability is expected to be higher. I urge the authors to clarify this, indicating how have they consider their replicates in each experiment. Additionally, the statistics should be performed on the data including all the independent technical – biological data points.

As examples:

In figure 1 all the qPCR plots show only representative technical replicates from 1 of the biological replicates mentioned in the text. All the data points from all the experiments should be displayed, with at least 3 independent experiments for each of the two biological replicates mentioned in the legend.

In figure 2, it is still unclear from the legends if the data points shown are from different technical and biological replicates or not.

In figure 3, I understand that in panels k, l and n are not from independent technical/biological replicates. The authors should repeat this experiment independently in order to prove their point, especially due to the small effect size observed in l and n

Figure 4h shows the data points from 1 representative experiment out of the three biological replicates performed.

Again, in all cases the authors should include all the data points and perform the statistics on the full datasets. This is particularly important due to the small effect size observed by the effect of Amuc14009. The same applies for all the supplementary figures.

2) The data supporting the Amuc14009-Ecadherin mechanism in organoids is still not convincing. As observed in figure 4g the deletion of E-cadherin alone already have a negative effect on the number of lobes per organoid. Additionally, there is no difference between the number of lobes in Cdh1f/f organoids independently of whether they have been treated or not with Amuc14009. Therefore, this experiment alone does not allow discerning if the effect observed is due to the interaction of Amuc14009 and E-cadherin or just from Ecad deletion alone. Additionally and as mentioned before, the authors should clarify the N number of biological/technical replicates. The authors should show that the differences are still significant after including/clarifying this point. As

remarked during the previous round of revision and earlier in this one, the fold change compared to control are minimal. Thus, despite being significant the authors should show that it has any functional downstream effect on the organoids. In my opinion, including data experimental data directly from the *Lgr5-CreERT2;Cdh1fl/fl* mice treated with 4-OHT and/or Amuc14009 (IHC or IF staining of several markers and qPCRs) could help supporting their claim. Additionally, does *ECadfl/fl* abrogate the effects shown in figures 2 and 3?

Minor points. Extended data figure 8c does not really show a differential E-cad internalization between the two conditions as claimed in the text. The authors should include an image where this effect is visible if possible or otherwise, modify their statement. The authors should include the data from Fig. R14 concerning the quantification of figure 4c in the manuscript.

We thank all three referees for the kind review of our manuscript. We carefully studied all comments from the reviewers and performed new experiments/analyses. We have therefore addressed the reviewers' comments and made corresponding changes in the revised manuscript. The following are our point-by-point responses to the reviewers' comments.

Reviewers Comments:

Reviewer #1 (Remarks to the Author):

This revised version of the manuscript on Amuc_1409 has been greatly improved. The authors also provided considerably new information that is now mainly in the detailed responses to the reviewers. They also removed some erroneous conclusions on the mucin induction.

However, a few points remain and new ones popped up:

Response: We sincerely thank you for your precious time in reviewing our manuscript and for providing insightful comments.

Q1. The amount of Amuc_1409 in intestinal *A.muciniphila* cells that is signaling-competent should be addresses more quantitatively: the protein when purified signals but how much is need for signal transduction ... do the authors see eignaling with whole cells? These important points should be also included in the manuscript like Fig R7. Here the cells are grown in basal medium – what then about mucin as this was used to grow the cells initially. I also suggest to include Fig R7 in the revised manuscript.

Response: We sincerely appreciate the reviewer's insightful suggestion. However, because the expression levels of Amuc_1409 in the cell-free culture supernatant were measured to be highly abundant, compared to those in other subcellular locations as described in our Response to Q2, we firmly believe that examining signalling from the cell-free culture supernatant, which contains the same effective dose of purified Amuc_1409, rather than whole cells, is the more appropriate approach. Based on our previous round of revision data (Fig. R7, for review purposes only), we have quantified the amount of Amuc_1409 in the cell-free culture

supernatants of *A. muciniphila* grown under basal medium, and the volumes of culture supernatant corresponding to different concentration of Amuc_1409 (2, 4, 8 nM) were calculated. To minimize the treatment volume of supernatant to mIOs, we concentrated the supernatant 14-fold using a 3K cut-off Amicon centrifugal filter (Merck Millipore, Burlington, USA) after filtration using a 50K cut-off Amicon centrifugal filter (Merck Millipore). Then we treated *A. muiniphila* cell-free supernatant with mIOs, as described in the following table (Supplementary Table R1, for review purposes only), and evaluated their effect on the growth of mIOs. We found that treatment of *A. muciniphila* cell-free supernatant, although not achieving the efficacy of purified Amuc_1409*, promoted mIOs growth and increased the number of lobes per mIO in a dose-dependent manner (Fig. R1, for review purposes only). These results suggest that Amuc_1409, as a major bioactive factor secreted by *A. muciniphila*, play a key role in improving ISC regenerative function.

Unfortunately, measuring the levels of Amuc_1409 in intestinal *A. muciniphila* cells *in vivo* is challenging by existing methods due to technical difficulties and limitations. We agree that it is important to clarify the precise levels of Amuc_1409 produced by *A. muciniphila* in the gut or, if it exists, in circulating form under both healthy and diseased *in vivo* situations. Additionally determining whether similar effects can be observed with more physiologically relevant concentrations would contribute to a better understanding of the mode of action of *A. muciniphila* from a microbiological perspective. Therefore, we have incorporated these limitations of our present study into the discussion part of the revised manuscript (page 20; line 421–424).

Supplementary Table R1. Treatment volume of *A. muciniphila* cell-free supernatant containing each Amuc_1409 dose in mouse small intestinal organoids

Amuc_1409 (nM)	1409 $\mu\text{g}/0.5 \text{ mL media}$	Volume of A. muciniphila cell-free supernatant containing each dose (μL)	Treatment volume of 14-fold concentrated supernatant (μL)
2	0.028	37.33	2.67
4	0.056	74.67	5.33
8	0.112	149.33	10.67

Fig. R1 Amuc_1409 promotes intestinal organoid growth as a major bioactive factor in *A. muciniphila* cell-free supernatant. a–d, Analysis of mIOs treated with PBS, medium only, Amuc_1409*, and dose-dependently 14-fold concentrated *A. muciniphila* cell-free supernatant (Sup.1: 2.67 μL , Sup.2: 5.33 μL , and Sup3: 10.67 μL) on day 4 after the second subculturing passage. Representative brightfield images of the mIOs (a), quantitative assessment of the percentage of budding organoids (b), the number of lobes per mIO (c), and the percentage distribution of organoids with the indicated number of lobes per mIO (d) in mIOs. Scale bar, 200 μm . Data are presented as the mean \pm SEM ($n = 2$ biological replicates). Statistical analyses

were performed using one-way ANOVA with Dunnett's multiple comparisons test ($*p < 0.05$, $**p < 0.01$, and $***p < 0.001$ vs PBS group).

Q2. The authors suggest that the Amuc_1409 protein is localized in the periplasm and outer membrane and conclude this from Fig R2 (see below). This is not correct as in this Fig the amounts of loaded material are not provided – so the Amuc_1409 protein may be present but in homeopathic amounts (like in the OM) – this should be addressed and conclusions should be supported by quantitative data – mass spectroscopy based proteome analysis is a better way to quantify

Response: We thank the reviewer for pointing this out. In the previous Fig R2, western blot analysis was conducted on whole cell (WC), periplasmic (PP), and outer membrane (OM) fractions, loaded with 5 μ g protein per fraction sample. As the reviewer suggested, for a more accurate quantification of Amuc_1409 in each subcellular fraction, we investigated its subcellular localization by a mass spectrometry-based proteomic approach. The WC, PP fraction, OM fraction and cell-free culture supernatant were analyzed by nLC-MS/MS. From LFQ-based quantitative analysis, it showed that the expression levels of Amuc_1409 in the cell-free culture supernatant were measured to be highly abundant, compared to those in other subcellular locations. To further verify this LFQ-based approach, quantitative profiling of three proteins such as Amuc_0824 (PP protein), Amuc_1100 (OM protein), and Amuc_0198 (cytoplasmic protein), previously shown to localize to each compartment, was performed to measure the expression levels of these proteins in each subcellular location. Eventually, we found that Amuc_0824, Amuc_1100, and Amuc_0198 were predominantly identified in the PP

fraction, OM fraction, and WC, respectively. When comparing the abundance of these proteins, Amuc_1409 was found to localize in the WC, PP fraction, and OM fraction, but in homoeopathic amounts. These results support our suggestion that Amuc_1409 is one of the highly abundant secreted proteins in the culture supernatant. We have now included this new data and relevant information in Supplementary Fig. 8 of the revised manuscript (page 17; line 353–363).

Q3. The authors provide data on thermal stability and these are very useful – I suggest to include this in the manuscript as highly relevant.

Response: As the reviewer suggested, we have now included the previous Fig R18, related to thermal stability, and relevant information in Supplementary Fig. 9 of the revised manuscript (page 18; line 374–375).

Q4. The authors now show the Amuc_1409 protein to be a dimer – is dimerization needed for activity ?

Response: We basically agree that this is an important consideration. In the previous Fig R8, gel-filtration chromatography results of purified Amuc_1409 protein produced in *E.coli* revealed a predominant single peak corresponding to the fractions with homodimer without a minor peak of the monomer or different oligomeric states. This result strongly suggests that Amuc_1409 will be physiologically secreted as a homodimer form to the extracellular milieu. Therefore we think that homodimeric form of Amuc_1409 would show its effects on regulating ISC-mediated intestinal homeostasis through an E-cadherin dependently. Although we could

not determine whether the monomeric form of Amuc_1409 is sufficient for binding to E-cadherin and activating β -catenin signalling or whether its binding affinity to E-cadherin is influenced by its oligomeric state, these points would be intriguing avens for future research.

Q5. Do the authors have any data on the Amuc_1409 stability ? I liked the experiment of Fig R18 and suggest to include this in the revised manuscript. However, the revised title is misleading as all experiments are in mice...so human can be deleted from the new title.

Response: We appreciate the reviewer for pointing this out. The multiple aspects of Amuc_1409 stability for potential therapeutic use are currently under investigation. As the reviewer suggested, we have now included the data of the previous Fig R18 and relevant information in Supplementary Fig. 12 of the revised manuscript (page 19; line 412–414) and have removed the term ‘human’ and included this information in the title of the revised manuscript (page 1; line 1–2).

Reviewers Comments:

Reviewer #2 (Remarks to the Author):

The revised manuscript goes a good way in addressing the issues raised during the initial submission, and several aspects of the data are now much better controlled and convincing.

Response: We sincerely thank the reviewer for taking the time to review our manuscript and for providing constructive feedback.

Regarding the revised manuscript:

Q1. Staining for secretory cell markers and qPCR (ED Fig. 2) is convincing in showing that Amuc_1409 drives secretory cell differentiation, but there remains no targeted analysis of enterocyte differentiation. Muc3 (also called Muc17) is a marker of mature enterocytes and is induced by treatment, indicating that Amuc_1409 drives general rather than targeted differentiation but this requires validation by examination of other mature enterocyte marker genes (see PMID: 29144463 for options).

Response: We sincerely appreciate the reviewer's insightful suggestion. As the reviewer commented, we examined gene expression levels of *Fabp1*, *Fabp2*, *Apoc3*, and *Alpi* as mature enterocyte markers^{1,2} in Amuc_1409*-treated mIOs by qRT-PCR. Our findings indicate that Amuc_1409* treatment significantly upregulated the gene expression of markers for mature enterocytes in mIOs, suggesting Amuc_1409 enhances the differentiation of various intestinal epithelial cells (IECs) rather than targeting specific IECs. We have now included this new data and relevant information in Supplementary Fig. 1h of the revised manuscript (page 9; line 166–169).

Q2. Data concerning the interaction of Amuc_1409 and E-cadherin (Fig. 4) is now much more convincing; however the current data only supports a role for in Amuc_1409 in driving E-cadherin dependent changes in stem cells and proliferation. Differentiation markers should also be examined (qPCR, IF) to demonstrate that the claimed effects on differentiation are also lost in E-cadherin KO organoids.

Response: Thank you for pointing this out. We have included the new data and information in the Supplementary Fig. 7 of the revised manuscript (page 15; line 318–320), indicating that Amuc_1409* had no effect on the increase in relative mRNA expression for various differentiated IECs markers in 4-OHT-treated *Lgr5-CreERT2;Cdh1^{fl/fl}* mIOs.

Q3. Control data using different Amuc_1409 constructs (Fig. R13) should be incorporated into the manuscript.

Response: As the reviewer suggested, we have now incorporated the data of the previous Fig R13 and relevant information in Supplementary Fig. 13 of the revised manuscript (page 20; line 425–440).

Reviewer #3 (Remarks to the Author):

The authors have addressed a number of my comments. However, there are still some major remarks that in my opinion need clarification. Without this clarification, it is not possible to judge the validity of a number of conclusions obtained.

Response: We sincerely thank you for your precious time in reviewing our manuscript and for providing valuable comments.

Major remarks

Q1. The authors answered to my Q1 that they have included information regarding experimental N numbers and reproducibility. However, it is still not clear to me looking at the majority of the figures/panels how the data points presented represent technical or biological replicates, and whether these are from independent experiments. Additionally, the authors should define their consideration of technical and biological replicates, in order to have a proper view on the sufficiency of the data included here. Are technical replicates from independent experiments and biological replicates from organoid lines derived from different mice/humans? Or technical replicates are the same organoid sample measured 3 times and biological replicate independent experiment with the same organoid line? Are technical replicates measured at different organoid passages? Or the experiments are performed with several wells from same passage, simultaneously, pooling some wells together and using that as replicate? Can the authors clarify the meaning of “RNA pooled from n=12 individual wells per group” and how it relates to the replicates? Is a technical replicate the same RNA sample from these 12 pooled wells ran in 3 different wells of the qPCR plate? I think it is important to clarify the N number in all the plots shown in the manuscript due to the small effect size on mRNA level observed

along the manuscript and the small variability (depicted as SEM) observed between replicates. In organoids, especially from different biological replicates/donors the variability is expected to be higher. I urge the authors to clarify this, indicating how have they consider their replicates in each experiment. Additionally, the statistics should be performed on the data including all the independent technical – biological data points.

As examples:

In figure 1 all the qPCR plots show only representative technical replicates from 1 of the biological replicates mentioned in the text. All the data points from all the experiments should be displayed, with at least 3 independent experiments for each of the two biological replicates mentioned in the legend.

In figure 2, it is still unclear from the legends if the data points shown are from different technical and biological replicates or not.

In figure 3, I understand that in panels k, l and n are not from independent technical/biological replicates. The authors should repeat this experiment independently in order to prove their point, especially due to the small effect size observed in l and n

Figure 4h shows the data points from 1 representative experiment out of the three biological replicates performed.

Again, in all cases the authors should include all the data points and perform the statistics on the full datasets. This is particularly important due to the small effect size observed by the effect of Amuc14009. The same applies for all the supplementary figures.

Response: We appreciate the reviewer for pointing this out. We have corrected information regarding experimental N numbers and reproducibility (technical or biological replicates) in all figure legends of the revised manuscript. In the *ex vivo* organoid assay, we define biological replicate and technical replicate as follows: Biological replicates are experiments conducted using independently established organoid lines derived from distinct mouse litter (for mIOs) or human embryonic stem cell (hESC) and human induced pluripotent stem cell (hiPSC) lines (for hIOs). In the qRT-PCR analysis of organoid samples, each biological replicate includes two or three technical replicates, meaning that the same RNA sample, pooled from individual cultured wells per group, was run in two or three different wells of the qRT-PCR plate.

During the revision period, we further repeated the mouse and human organoids experiment twice using independently established organoid lines derived from different two mice and two human stem cell lines (H1 hESC line, CRL2097 hiPSC line). Therefore, all mouse and human organoid experiments were biologically replicated three times and all replicates showed consistent reproducibility. We have included all the data points obtained from all the independent biological-technical replicates in the graphs and performed the statistics on the full data sets in the revised manuscript. In the graph of organoid experiments, each biological replicate is indicated by a different symbol. In the graph of *in vivo* experiments, each data point represents a biological replicate, corresponding to an individual mouse.

Q2. The data supporting the Amuc14009-Ecadherin mechanism in organoids is still not convincing. As observed in figure 4g the deletion of E-cadherin alone already have a negative effect on the number of lobes per organoid. Additionally, there is no difference between the

number of lobes in *Cdh1^{fl/fl}* organoids independently of whether they have been treated or not with Amuc14009. Therefore, this experiment alone does not allow discerning if the effect observed is due to the interaction of Amuc14009 and E-cadherin or just from *Ecad* deletion alone. Additionally and as mentioned before, the authors should clarify the N number of biological/technical replicates. The authors should show that the differences are still significant after including/clarifying this point. As remarked during the previous round of revision and earlier in this one, the fold change compared to control are minimal. Thus, despite being significant the authors should show that it has any functional downstream effect on the organoids. In my opinion, including data experimental data directly from the *Lgr5-CreERT2;Cdh1^{fl/fl}* mice treated with 4-OHT and/or Amuc14009 (IHC or IF staining of several markers and qPCRs) could help supporting their claim. Additionally, does *ECad^{fl/fl}* abrogate the effects shown in figures 2 and 3?

Response: As described in our Response to Q1, during the revision period, we further repeated the mouse organoids experiment twice using independently established organoid lines derived from two different *Lgr5-CreERT2;Cdh1^{fl/fl}* mice. Based on the full data sets obtained from all independently three biological replicates, we still observed a significant increase in the mIO growth of the Amuc_1409*-treated organoids in the absence of 4-OHT treatment when compared to that of the control organoids. In contrast, the deletion of E-cadherin in ISCs abrogated the effects of Amuc_1409* on mIO growth. The reviewer has raised important points regarding the negative effect of E-cadherin deletion on the lobe number per organoid. However, we believe that there are no major issues in using 4-OHT-treated mIOs from *Lgr5-creERT2;Cdh1^{fl/fl}* mice to investigate whether E-cadherin-dependent signalling is required for

the Amuc_1409-mediated enhancement of ISC function, because 4-OHT-treated mIOs from *Lgr5-creERT2;Cdh1^{fl/fl}* mice, despite showing a reduction in the number of lobes, did not exhibit a phenotype of not being able to grow or form crypt structures. Furthermore, we think that if the effect of Amuc_1409 on ISC function is not dependent on E-cadherin, Amuc_1409 would have increased the lobe number in 4-OHT-treated mIOs, regardless of the negative effect resulting from E-cadherin deletion.

As suggested by another reviewer, we have included new data in Supplementary Fig. 7 of the revised manuscript (page 15; line 318–320), indicating that Amuc_1409* had no effect on the increase in relative mRNA expression of various differentiated IECs markers in 4-OHT-treated *Lgr5-CreERT2;Cdh1^{fl/fl}* mIOs. Additionally, concerning the organoid experiments derived from *Lgr5-creERT2;Cdh1^{fl/fl}* mice, we have included all the data points obtained from all the independent biological-technical replicates in the graphs, clarified the N number of biological/technical replicates in the revised figure legends, and performed the statistics on the full data sets in the revised manuscript. Therefore, we strongly believe that the present study has convincingly demonstrated the indispensable role of E-cadherin-dependent signalling in the Amuc_1409-mediated improvement of ISC regenerative capacity by providing additional data from three independent biological replicates in the revised manuscript.

Q3. Minor points. Extended data figure 8c does not really show a differential E-cad internalization between the two conditions as claimed in the text. The authors should include an image where this effect is visible if possible or otherwise, modify their statement. The

authors should include the data from Fig. R14 concerning the quantification of figure 4c in the manuscript.

Response: We appreciate the reviewer for pointing this out. As the reviewer suggested, we have changed the IF images for E-cadherin in Supplementary Fig. 6c of the revised manuscript, clearly showing the effect of Amuc_1409 on E-cadherin internalization. Additionally, we have now included the quantification data of revised Fig. 6c (previous Fig R14) and relevant information in Fig. 6d of the revised manuscript (page 14; line 291–294).

References

1. Haber, A.L., *et al.* A single-cell survey of the small intestinal epithelium. *Nature* **551**, 333-339 (2017).
2. Baulies, A., *et al.* The Transcription Co-Repressors MTG8 and MTG16 Regulate Exit of Intestinal Stem Cells From Their Niche and Differentiation Into Enterocyte vs Secretory Lineages. *Gastroenterology* **159**, 1328-1341 e1323 (2020).

REVIEWERS' COMMENTS

Reviewer #2 (Remarks to the Author):

The authors have fully addressed my comments from the last round of review at Nature Microbiology and I have no additional points to raise, apart from a comment that there is space in the final manuscript I think it is important that the information concerning the finding that Amuc_1409 drives general epithelial differentiation should be included in the main figures.

Reviewer #3 (Remarks to the Author):

The authors have addressed my comments convincingly, and the revised version of the manuscript has improved significantly.

Only a minor remark remains:

-In figure 7a the label color and the conditions seem to have been mislabeled.

Reviewers Comments:

Reviewer #2 (Remarks to the Author):

The authors have fully addressed my comments from the last round of review at Nature Microbiology and I have no additional points to raise, apart from a comment that there is space in the final manuscript I think it is important that the information concerning the finding that Amuc_1409 drives general epithelial differentiation should be included in the main figures.

Response: We appreciate the reviewer's comment. Following the reviewer's suggestion, we have now incorporated the data from the previous Supplementary Figure 1, demonstrating that Amuc_1409 promotes general epithelial differentiation, into the main Figure 2 of the revised manuscript.

Reviewer #3 (Remarks to the Author):

The authors have addressed my comments convincingly, and the revised version of the manuscript has improved significantly.

Only a minor remark remains:

Q1. In figure 7a the label color and the conditions seem to have been mislabeled.

Response: We thank the reviewer for bringing to our attention the mislabeled legend color. We have rectified this in Figure 8a of the revised manuscript.